# On a continuous time model of gradient descent dynamics and instability in deep learning

**Mihaela Rosca**                                              *mihaelacr@deepmind.com*
*DeepMind, University College London*

**Yan Wu**                                                          *yanwu@deepmind.com*
*DeepMind*

**Chongli Qin**                                               *chongliqin@deepmind.com*
*DeepMind*

**Benoit Dherin**                                                  *dherin@google.com*
*Google*

**Reviewed on OpenReview:** *https://openreview.net/forum?id=EYrRzKPinA*

## Abstract

The recipe behind the success of deep learning has been the combination of neural networks and gradient-based optimization. Understanding the behavior of gradient descent however, and particularly its instability, has lagged behind its empirical success. To add to the theoretical tools available to study gradient descent we propose *the principal flow* (PF), a continuous time flow that approximates gradient descent dynamics. To our knowledge, the PF is the only continuous flow that captures the divergent and oscillatory behaviors of gradient descent, including escaping local minima and saddle points. Through its dependence on the eigendecomposition of the Hessian the PF sheds light on the recently observed edge of stability phenomena in deep learning. Using our new understanding of instability we propose a learning rate adaptation method which enables us to control the trade-off between training stability and test set evaluation performance.

## 1 Introduction

Our goal is to use continuous time models to understand the behavior of gradient descent. Using continuous dynamics to understand discrete time systems opens up tools from dynamical systems such as stability analysis, and has a long history in optimization and machine learning (Glendinning, 1994; Saxe et al., 2013; Nagarajan and Kolter, 2017; Lampinen and Ganguli, 2018; Arora et al., 2018; Advani et al., 2020; Elkabetz and Cohen, 2021; Vardi and Shamir, 2021; Franca et al., 2020; Barrett and Dherin, 2021; Smith et al., 2021). Most theoretical analysis of gradient descent using continuous time systems uses the negative gradient flow, but this has well known limitations such as not being able to explain any behavior contingent on the learning rate. To mitigate these limitations we find a new continuous time flow which reveals important new roles of the Hessian in gradient descent training. To do so, we use backward error analysis (BEA), a method with a long history in the numerical integration community (Hairer et al., 2006) that has only recently been used in the deep learning context (Barrett and Dherin, 2021; Smith et al., 2021).

We find that the proposed flow sheds new light on gradient descent stability, including but not limited to divergent and oscillatory behavior around a fixed point. Instability — areas of training where the loss consistently increases — and edge of stability behaviors (Cohen et al., 2021) —areas of training where the loss does not behave monotonically but decreases over long time periods — are pervasive in deep learning and occur for all learning rates and architectures Cohen et al. (2021); Gur-Ari et al. (2018); Gilmer et al. (2021); Lewkowycz et al. (2020). We use our novel insights to understand and mitigate these instabilities.

The structure of the presented work is as follows:

- We discuss the advantages of a continuous time approach in Section 2, where we also highlight the limitations of existing continuous time flows.

- We introduce **the principal flow** (the PF), a flow in complex space defined by the eigendecomposition of the Hessian (Section 3). To our knowledge the PF is the first continuous time flow that captures that gradient descent can diverge around local minima and saddle points. We show that using a complex flow is crucial in understanding instabilities in gradient descent.

- We show the PF is better than existing flows at modelling neural network training dynamics in Section 4. In Section 5 we use the PF to shed new light on edge of stability behaviors in deep learning. We do so by connecting changes in the loss and Hessian eigenvalues with core quantities exposed by the PF and neural network landscapes explored through the behavior of gradient flows.

- Through a continuous time perspective we demonstrate empirically how to control the trade-off between stability and performance in deep learning in Section 6. We do so using DAL (Drift Adjusted Learning rate), an approach to setting the learning rate dynamically based on insights on instability derived from the PF.

- We end by showcasing the potential of integrating our continuous time approach with other optimization schemes and highlighting how the PF can be used as a tool for existing continuous time analyses in Section 7.

**Notation**: We denote as $E$ the loss function, $\boldsymbol{\theta}$ the parameter vector of dimension $D$, $\nabla_{\boldsymbol{\theta}}^2 E$ the loss Hessian and $\lambda_i$ the Hessian's $i$'th largest eigenvalue with $\mathbf{u}_i$ the corresponding eigenvector. Since if $\mathbf{u}_i$ is an eigenvector of $\nabla_{\boldsymbol{\theta}}^2 E$ so is $-\mathbf{u}_i$, we always use $\mathbf{u}_i$ such that $Re[\nabla_{\boldsymbol{\theta}} E^T \mathbf{u}_i] \geq 0$; this has no effect on our results and is only used for convenience. For a continuous time flow $\boldsymbol{\theta}(h)$ refers to the solution of the flow at time $h$.

**Experiments**: A list of figures with details on how to reproduce each of them is provided in the Appendix. Code available at `https://github.com/deepmind/discretisation_drift`.

## 2 Continuous time models of gradient descent

The aim of this work is to understand the dynamics of gradient descent updates with learning rate $h$

$$\boldsymbol{\theta}_t = \boldsymbol{\theta}_{t-1} - h\nabla_{\boldsymbol{\theta}} E(\boldsymbol{\theta}_{t-1}) \tag{1}$$

from the perspective of continuous dynamics. When using continuous time dynamics to understand gradient descent it is most common to use *the negative gradient flow* (NGF)

$$\dot{\boldsymbol{\theta}} = -\nabla_{\boldsymbol{\theta}} E \tag{2}$$

Gradient descent can be obtained from the NGF through Euler numerical integration, with an error of $\mathcal{O}(h^2)$ after one gradient descent step. Studying gradient descent and its behavior around equilibria and beyond has thus taken two main approaches: directly studying the discrete updates of Eq 1 (Bartlett et al., 2018a;b; Mescheder et al., 2017; Gunasekar et al., 2018; Du et al., 2019; Allen-Zhu et al., 2019; Du and Hu, 2019; Ziyin et al., 2021a; Liu et al., 2021), or the continuous time NGF of Eq 2 (Glendinning, 1994; Saxe et al., 2013; Nagarajan and Kolter, 2017; Lampinen and Ganguli, 2018; Arora et al., 2018; Advani et al., 2020; Elkabetz and Cohen, 2021; Vardi and Shamir, 2021; Franca et al., 2020; Balduzzi et al., 2018). The appeal of continuous time systems lies in their connection with dynamical systems and the plethora of tools that thus become available, such as stability analysis; the simplicity by which conserved quantities can be obtained (Du et al., 2018; Franca et al., 2020); and analogies that can be constructed through similarities with physical systems (Franca et al., 2020). Because of the availability of tools for the analysis of continuous time systems, it has been previously noted that discrete time approaches are often more challenging and discrete time proofs are often inspired from continuous time ones (May, 1976; Elkabetz and Cohen, 2021). We use an example to showcase the ease of continuous time analyses: when following the NGF the loss $E$ decreases since $\frac{dE}{dt} = \frac{dE}{d\boldsymbol{\theta}}^T \frac{d\boldsymbol{\theta}}{dt} = -||\nabla_{\boldsymbol{\theta}} E||^2$. Showing that and *when* following the discrete time gradient

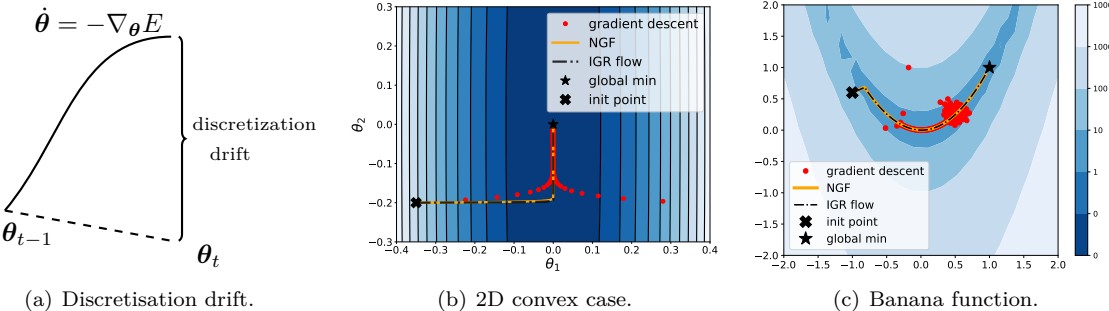

(a) Discretisation drift.  (b) 2D convex case.  (c) Banana function.

Figure 1: **Motivation**. Using continuous time flows to understand gradient descent is limited by the gap between the discrete and continuous dynamics. In the case of the negative gradient flow, we call this gap *discretization drift*. Other flows have been introduced to capture part of the drift, but they also fail to capture the oscillatory or unstable behavior of gradient descent.

descent update in Eq 1 is more challenging and requires adapting the analysis on the form of the loss function $E$. Classical convergence guarantees associated with other optimization approaches such as natural gradient are also derived in continuous time (Amari, 1998; Ollivier, 2015a;b). By analyzing the properties of continuous time systems one can also determine whether optimizers should more closely follow the underlying continuous time flow (Song et al., 2018; Qin et al., 2020), what regularizers should be constructed to ensure convergence or stability (Nagarajan and Kolter, 2017; Balduzzi et al., 2018; Rosca et al., 2021), construct converge guarantees in functional space for infinitely wide networks (Jacot et al., 2018; Lee et al., 2019).

## 2.1 Limitations of existing continuous time flows

The well-known discrepancy between Euler integration and the NGF, often called *discretization error* or *discretization drift* (Figure 1(a)) leads to certain limitations when using the NGF to describe gradient descent, namely: the NGF cannot explain divergence around a local minima for high learning rates or convergence to flat minima as often seen in the training of neural networks. Critically, since the NGF does not depend on the learning rate, it cannot explain any learning rate dependent behavior.

The appeal of continuous time methods together with the limitations of the NGF have inspired the machine learning community to look for other continuous time systems which may better approximate the gradient descent trajectory. One approach to constructing continuous time flows approximating gradient descent that takes into account the learning rate is backward error analysis (BEA). Using this approach, Barrett and Dherin (2021) introduce the Implicit Gradient Regularization flow (IGR flow):

$$\dot{\boldsymbol{\theta}} = -\nabla_{\boldsymbol{\theta}} E - \frac{h}{2} \nabla_{\boldsymbol{\theta}}^2 E \nabla_{\boldsymbol{\theta}} E \tag{3}$$

which tracks the dynamics of the gradient descent step $\boldsymbol{\theta}_t = \boldsymbol{\theta}_{t-1} - h\nabla_{\boldsymbol{\theta}} E(\boldsymbol{\theta}_{t-1})$ with an error of $\mathcal{O}(h^3)$, thus reducing the order of the error compared to the NGF. Unlike the NGF flow, the IGR flow depends on the learning rate $h$. This dependence explains certain properties of gradient descent, such as avoiding trajectories with high gradient norm; the authors connect this behavior to convergence to flat minima.

Like the NGF flow however, the IGR flow does not explain the instabilities of gradient descent, as we illustrate in Figure 1. Indeed, Barrett and Dherin (2021) (their Remark 3.4) show that performing stability analysis around local minima using the IGR flow does not lead to qualitatively different conclusions from those using the NGF: both NGF and the IGR flow predict gradient descent to be always locally attractive around a local minimum (proofs in Section A.4.1), contradicting the empirically observed behavior of gradient descent. To understand why both the NFG and the IGR flow cannot capture oscillations and divergence around a local minimum, we note that stationary points $\nabla_{\boldsymbol{\theta}} E = \mathbf{0}$ are fixed points for both flows. We visualize an example in Figure 2(a): since to go from the initial point to the gradient descent iterates requires passing through the local minimum, both flows would stop at the local minimum and never reach the following gradient descent

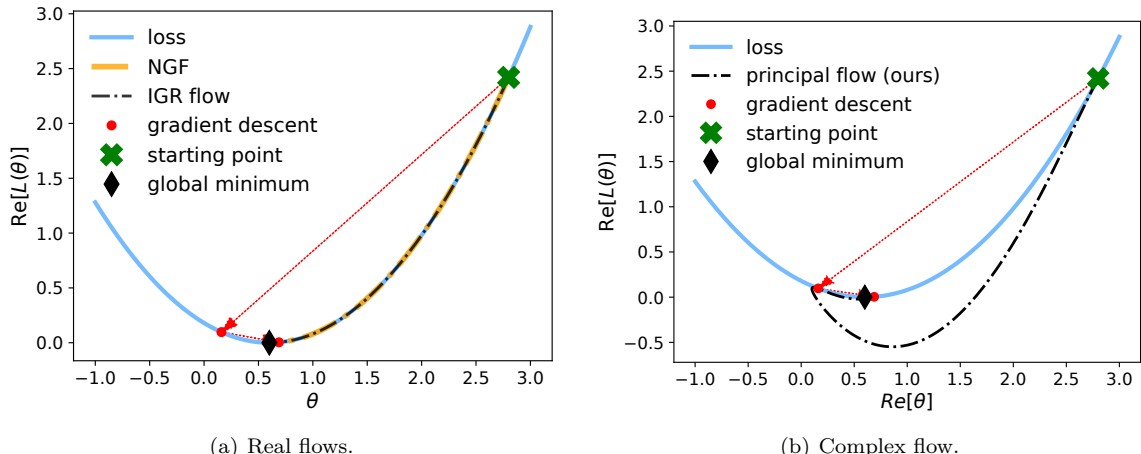

(a) Real flows.             (b) Complex flow.

Figure 2: **Complex flows capture oscillations and divergence around local minima**. In the real space, the trajectory going from the starting point to the second gradient descent iterate goes through the global minima, and real flows stop there. In complex space however, that need not be the case.

iterates. In the case of neural networks we show in Figure 31 in the Appendix that while the IGR flow is better than the NGF at describing gradient descent, a substantial gap remains.

The lack of ability of existing continuous time flows to model instabilities empirically observed in gradient descent such as those shown in Figure 1 has been used as a motivation to use discrete-time methods instead (Yaida, 2018; Liu et al., 2021). The goal of our work is to overcome this issue by introducing a novel continuous time flow which captures instabilities observed in gradient descent. To do so, we follow the footsteps of Barrett and Dherin (2021) and use Backward Error Analysis. By using a continuous time flow we can leverage the tools and advantages of continuous time methods discussed earlier in this section; by incorporating discretization drift into our model of gradient descent we can increase their applicability to explain unstable training behavior. Indeed, we show in Figure 2(b) that the flow we propose captures the training instabilities; a key reason why is that, unlike existing flows, it operates in complex space. In Section 3 we show the importance of operating in complex space in order to understand oscillatory and instability behaviors of gradient descent.

## 2.2 Backward error analysis

Backward error analysis (BEA) is a tool in numerical analysis developed to understand the discretization error of numerical integrators. We now present an overview of how to use it in the context of gradient descent; for a general overview see Hairer et al. (2006). BEA provides a modified vector field:

$$\tilde{f}_n(\boldsymbol{\theta}) = -\nabla_{\boldsymbol{\theta}} E + h f_1(\boldsymbol{\theta}) + \cdots + h^n f_n(\boldsymbol{\theta}), \tag{4}$$

by finding functions $f_1, \dots f_n$ such that the solution of the modified ODE at order $n$, that is,

$$\dot{\tilde{\boldsymbol{\theta}}} = -\nabla_{\boldsymbol{\theta}} E + h f_1(\boldsymbol{\theta}) + \cdots + h^n f_n(\boldsymbol{\theta}) \tag{5}$$

follows the discrete dynamics of the gradient descent update with an error $\|\boldsymbol{\theta}_t - \tilde{\boldsymbol{\theta}}(h)\|$ of order $\mathcal{O}(h^{n+2})$, where $\tilde{\boldsymbol{\theta}}(h)$ is the solution of the modified equation truncated at order $n$ at time $h$, with $\tilde{\boldsymbol{\theta}}(0) = \boldsymbol{\theta}_{t-1}$. The full modified vector field with all orders $(n \to \infty)$

$$\tilde{f}(\boldsymbol{\theta}) = -\nabla_{\boldsymbol{\theta}} E + h f_1(\boldsymbol{\theta}) + \cdots + h^n f_n(\boldsymbol{\theta}) + \cdots, \tag{6}$$

is usually divergent and only forms an asymptotic expansion. What BEA provides is the Taylor expansion in $h$ of an unknown $h$-dependent vector field $f_h(\boldsymbol{\theta})$ developed at $h = 0$:

$$\tilde{f}(\boldsymbol{\theta}) = \text{Taylor}_{h=0} f_h(\boldsymbol{\theta}). \tag{7}$$

Thus a strategy for finding $f_h$ is to find a series of the form in Eq 6 via BEA and then find the function $f_h$ such that its Taylor expansion in $h$ at 0 results in the found series. Using this approach we can find the flow $\dot{\tilde{\boldsymbol{\theta}}} = f_h(\tilde{\boldsymbol{\theta}})$ which exactly describes the gradient descent step $\boldsymbol{\theta}_t = \boldsymbol{\theta}_{t-1} - h\nabla_{\boldsymbol{\theta}}E(\boldsymbol{\theta}_{t-1})$.

While flows obtained using BEA are constructed to approximate one gradient descent step, the same flows can be used over multiple gradient descent steps as shown in Section A.9 in the Appendix.

**BEA proofs**. The general structure of BEA proofs is as follows: start with a Taylor expansion in $h$ of the modified flow in Eq 5; write each term in the Taylor expansion as a function of $\nabla_{\boldsymbol{\theta}}E$ and the desired $f_i$ (this often requires applying the chain rule repeatedly); group together terms of the same order in $h$ in the expansion; and identify $f_i$ such that all terms of $\mathcal{O}(h^p)$ are 0 for $p \geq 2$, as is the case in the gradient descent update. A formal overview of BEA proofs can be found in Section A.1 in the Appendix.

We now exemplify how to use BEA to find the IGR flow (Eq 3) (Barrett and Dherin, 2021). Since we are only looking for the first correction term, we only need to find $f_1$. We perform a Taylor expansion to find the value of $\tilde{\boldsymbol{\theta}}(h)$ up to order $\mathcal{O}(h^3)$ and then identify $f_1$ from that expression such that the error $\|\boldsymbol{\theta}_t - \tilde{\boldsymbol{\theta}}(h)\|$ is of order $\mathcal{O}(h^3)$. We have: $\tilde{\boldsymbol{\theta}}(h) = \boldsymbol{\theta}_{t-1} + h\tilde{\boldsymbol{\theta}}^{(1)}(\boldsymbol{\theta}_{t-1}) + \frac{h^2}{2}\tilde{\boldsymbol{\theta}}^{(2)}(\boldsymbol{\theta}_{t-1}) + \mathcal{O}(h^3)$. We know by the definition of the modified vector field (Eq 5) that $\tilde{\boldsymbol{\theta}}^{(1)} = -\nabla_{\boldsymbol{\theta}}E + hf_1(\tilde{\boldsymbol{\theta}})$. We can then use the chain rule to obtain $\tilde{\boldsymbol{\theta}}^{(2)} = \frac{-\nabla_{\boldsymbol{\theta}}E + hf_1(\boldsymbol{\theta})}{dt} = \frac{-\nabla_{\boldsymbol{\theta}}E}{dt} + \mathcal{O}(h) = \frac{-\nabla_{\boldsymbol{\theta}}E}{d\boldsymbol{\theta}}\frac{d\boldsymbol{\theta}}{dt} + \mathcal{O}(h) = \nabla_{\boldsymbol{\theta}}^2 E\nabla_{\boldsymbol{\theta}}E + \mathcal{O}(h)$. Thus $\tilde{\boldsymbol{\theta}}(h) = \boldsymbol{\theta}_{t-1} - h\nabla_{\boldsymbol{\theta}}E(\boldsymbol{\theta}_{t-1}) + h^2 f_1(\boldsymbol{\theta}_{t-1}) + \frac{h^2}{2}\nabla_{\boldsymbol{\theta}}^2 E(\boldsymbol{\theta}_{t-1})\nabla_{\boldsymbol{\theta}}E(\boldsymbol{\theta}_{t-1}) + \mathcal{O}(h^3)$. We can then write $\boldsymbol{\theta}_t - \tilde{\boldsymbol{\theta}}(h) = \boldsymbol{\theta}_{t-1} - h\nabla_{\boldsymbol{\theta}}E(\boldsymbol{\theta}_{t-1}) - \left(\boldsymbol{\theta}_{t-1} - h\nabla_{\boldsymbol{\theta}}E(\boldsymbol{\theta}_{t-1}) + hf_1(\boldsymbol{\theta}_{t-1}) + \frac{h^2}{2}\nabla_{\boldsymbol{\theta}}^2 E(\boldsymbol{\theta}_{t-1})\nabla_{\boldsymbol{\theta}}E(\boldsymbol{\theta}_{t-1}) + \mathcal{O}(h^3)\right)$. After simplifying we obtain $\boldsymbol{\theta}_t - \tilde{\boldsymbol{\theta}}(h) = h^2 f_1(\boldsymbol{\theta}_{t-1}) + \frac{h^2}{2}\nabla_{\boldsymbol{\theta}}^2 E(\boldsymbol{\theta}_{t-1})\nabla_{\boldsymbol{\theta}}E(\boldsymbol{\theta}_{t-1}) + \mathcal{O}(h^3)$. For the error to be of order $\mathcal{O}(h^3)$ the terms of order $\mathcal{O}(h^2)$ have to be $\boldsymbol{0}$. This entails $f_1 = -\frac{1}{2}\nabla_{\boldsymbol{\theta}}^2 E\nabla_{\boldsymbol{\theta}}E$ leading to Eq 3.

## 3 The principal flow

In the previous section we have seen how BEA can be used to define continuous time flows which capture the dynamics of gradient descent up to a certain order in learning rate. We have also explored the limitations of these flows, including the lack of ability to explain oscillations observed empirically when using gradient descent. To further expand our understanding of gradient descent via continuous time methods, we would like to get an intuition for the structure of higher order modified vector fields provided by BEA. We start with the following modified vector field, which we will call *the third order flow* (proof in Section A.2):

$$\dot{\boldsymbol{\theta}} = -\nabla_{\boldsymbol{\theta}}E - \frac{h}{2}\nabla_{\boldsymbol{\theta}}^2 E\nabla_{\boldsymbol{\theta}}E - h^2\left(\frac{1}{3}(\nabla_{\boldsymbol{\theta}}^2 E)^2\nabla_{\boldsymbol{\theta}}E + \frac{1}{12}\nabla_{\boldsymbol{\theta}}E^T(\nabla_{\boldsymbol{\theta}}^3 E)\nabla_{\boldsymbol{\theta}}E\right) \tag{8}$$

The third order flow tracks the dynamics of the gradient descent step $\boldsymbol{\theta}_t = \boldsymbol{\theta}_{t-1} - h\nabla_{\boldsymbol{\theta}}E(\boldsymbol{\theta}_{t-1})$ with an error of $\mathcal{O}(h^4)$, thus further reducing the order of the error compared to the IGR flow. Like the IGR flow and the NGF, the third order flow has the property that $\dot{\boldsymbol{\theta}} = \boldsymbol{0}$ if $\nabla_{\boldsymbol{\theta}}E = \boldsymbol{0}$ and thus will exhibit the same limitations observed in Figure 2. The third order flow allows us to spot a pattern: the correction term of order $\mathcal{O}(h^n)$ in the BEA modified flow describing gradient descent contains the term $(\nabla_{\boldsymbol{\theta}}^2 E)^n\nabla_{\boldsymbol{\theta}}E$ and terms which contain higher order derivatives with respect to parameters, terms which we will denote as $\mathcal{C}(\nabla_{\boldsymbol{\theta}}^3 E)$.

**Our approach**. We will use the terms of the form $(\nabla_{\boldsymbol{\theta}}^2 E)^n\nabla_{\boldsymbol{\theta}}E$ to construct a new continuous time flow. We will take a three-step approach. First, for an arbitrary order $\mathcal{O}(h^n)$ we will find the terms containing only first and second order derivatives in the modified vector field given by BEA and show they are of the form $(\nabla_{\boldsymbol{\theta}}^2 E)^n\nabla_{\boldsymbol{\theta}}E$ (Theorem 3.1). Second, we will use all orders to create a series (Corollary 3.1). Third, we will use the series to find the modified flow given by BEA (Theorem 3.2). All proofs are provided in Section A of the Appendix.

**Theorem 3.1** *The modified vector field with an error of order $\mathcal{O}(h^{n+2})$ to the gradient descent update $\boldsymbol{\theta}_t = \boldsymbol{\theta}_{t-1} - h\nabla_{\boldsymbol{\theta}}E(\boldsymbol{\theta}_{t-1})$ has the form:*

$$\dot{\boldsymbol{\theta}} = \sum_{p=0}^{n}\frac{-1}{p+1}h^p(\nabla_{\boldsymbol{\theta}}^2 E)^p\nabla_{\boldsymbol{\theta}}E + \mathcal{C}(\nabla_{\boldsymbol{\theta}}^3 E) \tag{9}$$

*where $\mathcal{C}(\nabla^3_{\boldsymbol{\theta}}E)$ denotes the family of functions which can be written as a sum of terms, each term containing a derivative of higher order than 3 with respect to parameters.*

The result is proven by induction. The base cases for $n = 1, 2$ and 3 follow from the NGF, IGR and third order flows. For higher order terms, the proof uses induction to find the term in $f_i$ depending on $\nabla^2_{\boldsymbol{\theta}}E$ and $\nabla_{\boldsymbol{\theta}}E$ only and follows the BEA proof structure highlighted in Section 2.2, but Step 3 is modified to not account for terms in $\mathcal{C}(\nabla^3_{\boldsymbol{\theta}}E)$. From the above, we can obtain the following corollary by using all orders $n$ and the eigen decomposition of $\nabla^2_{\boldsymbol{\theta}}E$:

**Corollary 3.1** *The full order modified flow obtained by performing BEA on gradient descent updates is of the form:*

$$\dot{\boldsymbol{\theta}} = \sum_{p=0}^{\infty} \frac{-1}{p+1} h^p (\nabla^2_{\boldsymbol{\theta}}E)^p \nabla_{\boldsymbol{\theta}}E + \mathcal{C}(\nabla^3_{\boldsymbol{\theta}}E) = \sum_{p=0}^{\infty} \frac{-1}{p+1} h^p \left( \sum_{i=0}^{D-1} \lambda_i^p \mathbf{u}_i \mathbf{u}_i^T \right) \nabla_{\boldsymbol{\theta}}E + \mathcal{C}(\nabla^3_{\boldsymbol{\theta}}E) \tag{10}$$

$$= \sum_{i=0}^{D-1} \left( \sum_{p=0}^{\infty} \frac{-1}{p+1} h^p \lambda_i^p \right) (\nabla_{\boldsymbol{\theta}}E^T \mathbf{u}_i) \mathbf{u}_i + \mathcal{C}(\nabla^3_{\boldsymbol{\theta}}E) \tag{11}$$

*where $\lambda_i$ and $\mathbf{u}_i$ are the respective eigenvalues and eigenvectors of the Hessian $\nabla^2_{\boldsymbol{\theta}}E$.*

If $\lambda_0 > 1/h$ the BEA series above diverges. Generally BEA series are not convergent and approximate the discrete scheme only by truncation (Hairer et al., 2006). When the series in Eq 11 diverges, truncating it up to any order $n$ however will result in a flow which will not be able to capture instabilities, even in the quadratic case. Such flows (including the IGR flow) will always predict the loss function will decrease for a quadratic loss where a minimum exists, since: $\frac{dE}{dt} = \nabla_{\boldsymbol{\theta}}E^T \left( \sum_{p=0}^{n} \frac{-1}{p+1} h^p (\nabla^2_{\boldsymbol{\theta}}E)^p \nabla_{\boldsymbol{\theta}}E \right) = - \sum_{p=0}^{n} \frac{1}{p+1} h^p \sum_{i=0}^{D-1} (\lambda_i^p)(\nabla_{\boldsymbol{\theta}}E^T \mathbf{u}_i)^2$ which is never positive for any quadratic loss where a minimum exists (i.e. when $\lambda_i \geq 0, \forall i$). The above also entails that the flows always predict convergence around a local minimum, which is not the case for gradient descent which can diverge for large learning rates.

To further track instabilities we can use the BEA series to formulate the following flow:

**Definition 3.1** *We define the **principal flow** (PF) as*

$$\dot{\boldsymbol{\theta}} = \sum_{i=0}^{D-1} \frac{\log(1 - h\lambda_i)}{h\lambda_i} (\nabla_{\boldsymbol{\theta}}E^T \mathbf{u}_i) \mathbf{u}_i \tag{12}$$

We note that $\lim_{\lambda \to 0} \frac{\log(1-h\lambda)}{h\lambda} = -1$ and thus the PF is well defined when the Hessian $\nabla^2_{\boldsymbol{\theta}}E$ is not invertible. Unlike the NGF and the IGR flow, the modified vector field of the PF cannot be always written as the gradient of a loss function in $\mathbb{R}$, and can be complex valued.

**Theorem 3.2** *The Taylor expansion in $h$ at $h = 0$ of the PF vector field coincides with the series coming from the BEA of gradient descent (Eq 11).*

**Proof:** Using the Taylor expansion $\text{Taylor}_{z=0} \frac{\log(1-z)}{z} = \sum_{p=0}^{\infty} \frac{-1}{p+1} z^p$ we obtain:

$$\text{Taylor}_{h=0} \sum_{i=0}^{D-1} \frac{\log(1 - h\lambda_i)}{h\lambda_i} (\nabla_{\boldsymbol{\theta}}E^T \mathbf{u}_i) \mathbf{u}_i = \sum_{i=0}^{D-1} \left( \sum_{p=0}^{\infty} \frac{-1}{p+1} h^p \lambda_i^p \right) (\nabla_{\boldsymbol{\theta}}E^T \mathbf{u}_i) \mathbf{u}_i \tag{13}$$

$\square$

We have used BEA to find the flow that when Taylor expanded at $h = 0$ leads to the series in Eq 11. When the BEA series in Eq 11 converges, namely $\lambda_0 < 1/h$, the PF and the flow given by the BEA series are the same. When $\lambda_0 > 1/h$ however, the PF is complex and the BEA series diverges. While in this case any BEA truncated flow will not be able to track gradient descent closely, we show that for quadratic losses the PF will track gradient descent exactly, and that it is a good model of gradient descent around fixed points. We show examples of the PF tracking gradient descent exactly in the quadratic case in Figures 2(b) and 5.

| Negative Gradient Flow | IGR Flow | Principal Flow |
|---|---|---|
| $\dot{\boldsymbol{\theta}} = \sum_{i=0}^{D-1} -(\nabla_{\boldsymbol{\theta}} E^T \mathbf{u}_i)\mathbf{u}_i$ | $\dot{\boldsymbol{\theta}} = \sum_{i=0}^{D-1} -(1 + \frac{h}{2}\lambda_i)(\nabla_{\boldsymbol{\theta}} E^T \mathbf{u}_i)\mathbf{u}_i$ | $\dot{\boldsymbol{\theta}} = \sum_{i=0}^{D-1} \frac{\log(1-h\lambda_i)}{h\lambda_i}(\nabla_{\boldsymbol{\theta}} E^T \mathbf{u}_i)\mathbf{u}_i$ |
| $\alpha_{NGF}(h\lambda_i) = -1$ | $\alpha_{IGR}(h\lambda_i) = -(1 + \frac{h}{2}\lambda_i)$ | $\alpha_{PF}(h\lambda_i) = \frac{\log(1-h\lambda_i)}{h\lambda_i}$ |

Table 1: Understanding the differences between the flows discussed in terms of the eigendecomposition of the Hessian. All flows have the form $\dot{\boldsymbol{\theta}} = \sum_{i=0}^{D-1} \alpha(h\lambda_i)(\nabla_{\boldsymbol{\theta}} E^T \mathbf{u}_i)\mathbf{u}_i$ with different $\alpha$ summarized here.

**Remark 3.1** *For quadratic losses of the form $E = \frac{1}{2}\boldsymbol{\theta}^T \mathbf{A}\boldsymbol{\theta} + \mathbf{b}^T\boldsymbol{\theta}$, the PF captures gradient descent exactly. This case has been proven in Hairer et al. (2006). The solution of the PF can also be computed exactly in terms of the eigenvalues of $\nabla_{\boldsymbol{\theta}}^2 E$: $\boldsymbol{\theta}(t) = \sum_{i=0}^{D-1} e^{\frac{\log(1-h\lambda_i)}{h}t}\boldsymbol{\theta}_0^T \mathbf{u}_i \mathbf{u}_i + t\sum_{i=0}^{D-1} \frac{\log(1-h\lambda_i)}{h\lambda_i}b^T\mathbf{u}_i$.*

**Remark 3.2** *In a small enough neighborhood around a critical point (where higher order derivatives can be ignored) the PF can be used to describe gradient descent dynamics closely. We show this also using a linearization argument in Section A.5 in the Appendix.*

**Definition 3.2** *The terms $\mathcal{C}(\nabla_{\boldsymbol{\theta}}^3 E)$ are called **non-principal terms**. The term $\frac{1}{12}\nabla_{\boldsymbol{\theta}} E^T (\nabla_{\boldsymbol{\theta}}^3 E)\nabla_{\boldsymbol{\theta}} E$ in Eq 8 is a non-principal term (we will call this term non-principal third order term).*

**Definition 3.3** *We define the **principal flow with third order non principal term** as*

$$\dot{\boldsymbol{\theta}} = \sum_{i=0}^{D-1} \frac{\log(1-h\lambda_i)}{h\lambda_i}(\nabla_{\boldsymbol{\theta}} E^T \mathbf{u}_i)\mathbf{u}_i - \underbrace{\frac{h^2}{12}\nabla_{\boldsymbol{\theta}} E^T (\nabla_{\boldsymbol{\theta}}^3 E)\nabla_{\boldsymbol{\theta}} E}_{\text{third order non principal term}} \tag{14}$$

General theoretical bounds on the error between continuous time flows and gradient descent are challenging to construct in the case of a general parametrised $E(\boldsymbol{\theta})$ as the error will be determined by the shape of $E$. We know the conditions which determine when certain flows follow gradient descent exactly. The NGF and gradient descent will follow the same trajectory in areas where $\nabla_{\boldsymbol{\theta}}^2 E \nabla_{\boldsymbol{\theta}} E = 0$ (see Theorem 6.1) and thus $E$ has a constant gradient in time, since $\frac{d\nabla_{\boldsymbol{\theta}} E}{dt} = \nabla_{\boldsymbol{\theta}}^2 E \nabla_{\boldsymbol{\theta}} E$ under the NGF. The PF generalises the NGF, in that it follows the same trajectory as gradient descent not only for trajectories where $\nabla_{\boldsymbol{\theta}}^2 E \nabla_{\boldsymbol{\theta}} E = 0$, but also when $E$ is quadratic. Informally, we can state that the closer we are to these exact conditions, the more likely the flows are to capture the dynamics of gradient descent. Formally, bounds on the error between GD and NGF can be provided by the Fundamental Theorem (Theorem 10.6 in Wanner and Hairer (1996)) which has recently been adapted to a neural network parametrisation by Elkabetz and Cohen (2021); this bound depends on the magnitude of the smallest Hessian eigenvalue along the NGF trajectory. We hope that future work can expand the Fundamental Theorem such that error bounds between the PF and gradient descent can be constructed for deep neural networks. Here we take an empirical approach and show that although not exact outside the quadratic case the PF captures key features of the gradient descent dynamics in stable or unstable regions of training, around and outside critical points, for small examples or large neural networks.

### 3.1 The principal flow and the eigen decomposition of the Hessian

All flows considered here have the form form $\dot{\boldsymbol{\theta}} = \sum_{i=0}^{D-1} \alpha(h\lambda_i)(\nabla_{\boldsymbol{\theta}} E^T \mathbf{u}_i)\mathbf{u}_i$, where $\alpha$ is a function computing the corresponding coefficient; we will denote the one associated with each flow as $\alpha_{NGF}$, $\alpha_{IGR}$ and $\alpha_{PF}$ respectively. For a side-by-side comparison between the NGF, IGR flow and the PF as functions of the Hessian eigendecomposition see Table 1. Since $\nabla_{\boldsymbol{\theta}} E^T \mathbf{u}_i \geq 0$, the $\alpha$ function determines the sign of a modified vector field in the direction $\mathbf{u}_i$. For brevity it will be useful to define the coefficient of $\mathbf{u}_i$ in the vector field of the PF:

**Definition 3.4** *We call $sc_i = \frac{\log(1-h\lambda_i)}{h\lambda_i}(\nabla_{\boldsymbol{\theta}} E^T \mathbf{u}_i) = \alpha_{PF}(h\lambda_i)\nabla_{\boldsymbol{\theta}} E^T \mathbf{u}_i$ the **stability coefficient** for eigendirection i. $sign(sc_i) = sign(\alpha_{PF}(h\lambda_i))$.*

In order to understand the PF and how it is different from the NGF we explore the change in each eigendirection $\mathbf{u}_i$ and we perform case analysis on the relative value of the eigenvalues $\lambda_i$ and the learning rate $h$. To do so, we will compare $\alpha_{NGF}(h\lambda_i)$ and $\alpha_{PF}(h\lambda_i)$ since the sign of $\alpha_{NGF}(h\lambda_i)$ determines the direction which minimises $E$ given by $\mathbf{u}_i$. Since our goal is to understand the behavior of gradient descent, we perform

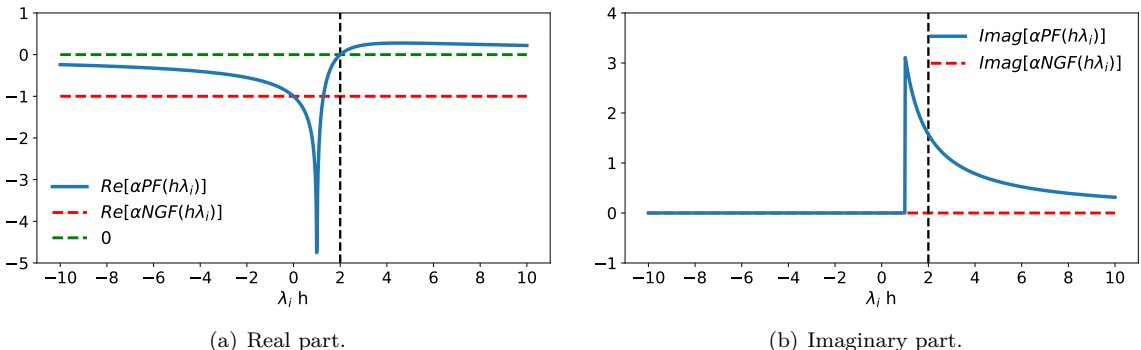

(a) Real part.                    (b) Imaginary part.

Figure 3: Comparing the coefficients $\alpha_{NGF}$ and $\alpha_{PF}$ across the training landscape.

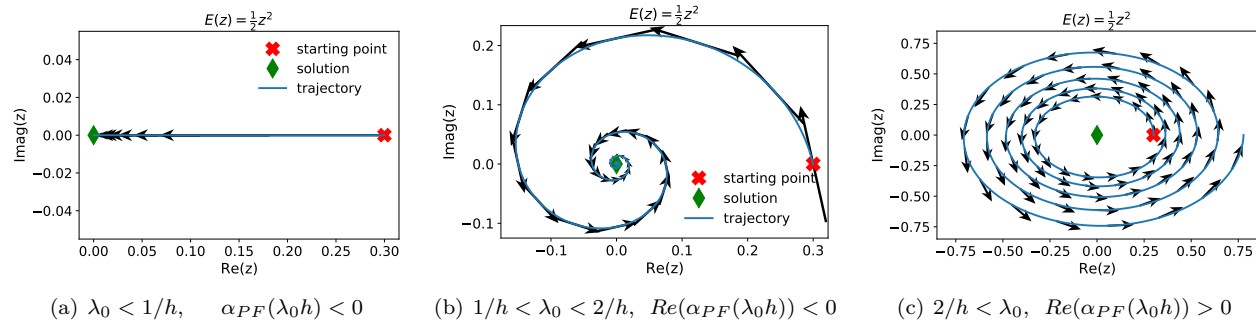

(a) $\lambda_0 < 1/h$,   $\alpha_{PF}(\lambda_0 h) < 0$   (b) $1/h < \lambda_0 < 2/h$,  $Re(\alpha_{PF}(\lambda_0 h)) < 0$   (c) $2/h < \lambda_0$,  $Re(\alpha_{PF}(\lambda_0 h)) > 0$

Figure 4: The behavior of PF on $E(z) = \frac{1}{2}z^2$ with solution $z(t) = e^{\log(1-h)/h}z(0)$. When $\lambda_0 < 1/h$, $z(t) = (1-h)^{t/h}z(0)$ which is in real space and converges to the equilibrium. When $\lambda_0 > 1/h$, $z(t) = (h-1)^{t/h}\left(\cos(\pi t/h) + i\sin(\pi t/h)\right)z(0)$. This exhibits oscillatory behavior, and when $\lambda_0 > 2/h$, diverges.

the case by case analysis of what happens at the start of a gradient descent iteration and thus use real values for $\lambda_i$ and $\mathbf{u}_i$ even when the PF is complex valued. We visualize $\alpha_{NGF}$ and $\alpha_{PF}$ in Figure 3 and we use Figure 4 to show examples of each case using a simple function.

**Real stable case**: $\lambda_i < 1/h$. $\text{sign}(\alpha_{NGF}(h\lambda_i)) = \text{sign}(\alpha_{PF}(h\lambda_i)) = -1$.

$\alpha_{NGF}(h\lambda_i) = -1$ and $\alpha_{PF}(h\lambda_i) = \frac{\log(1-h\lambda_i)}{h\lambda_i} < 0$. The coefficients of both the NGF and PF in eigendirection $\mathbf{u}_i$ are both negative and real. The case is exemplified in Figure 4(a).

**Complex stable case**: $1/h < \lambda_i < 2/h$. $\text{sign}(\alpha_{NGF}(h\lambda_i)) = \text{sign}(Re[\alpha_{PF}(h\lambda_i)]) = -1$. $\alpha_{PF}(h\lambda_i) \in \mathbb{C}$.

$\alpha_{NGF}(h\lambda_i) = -1$ and $\alpha_{PF}(h\lambda_i) = \frac{\log(1-h\lambda_i)}{h\lambda_i} = \frac{\log(-1+h\lambda_i)+i\pi}{h\lambda_i} \in \mathbb{C}$ and $Re[\alpha_{PF}(h\lambda_i)] = \frac{\log(-1+h\lambda_i)}{h\lambda_i} < 0$. The real part of the coefficient of the NGF and PF in eigendirection $\mathbf{u}_i$ are both negative. The imaginary part of $\alpha_{PF}$ can still introduce instability and oscillations, as we show in Figure 4(b).

**Unstable complex case**: $2/h < \lambda_i$. $\text{sign}(\alpha_{NGF}(h\lambda_i)) \neq \text{sign}(Re[\alpha_{PF}(h\lambda_i)])$. $\alpha_{PF}(h\lambda_i) \in \mathbb{C}$.

$\alpha_{NGF}(h\lambda_i) = -1$ and $\alpha_{PF}(h\lambda_i) = \frac{\log(1-h\lambda_i)}{h\lambda_i} = \frac{\log(-1+h\lambda_i)+i\pi}{h\lambda_i} \in \mathbb{C}$ and $Re[\alpha_{PF}(h\lambda_i)] = \frac{\log(-1+h\lambda_i)}{h\lambda_i} > 0$. The real part of the coefficient of the NGF in eigendirection $\mathbf{u}_i$ is negative, while the real part of the coefficient of the PF is positive. The PF goes in the opposite direction of the NGF which minimises E; this change in sign can cause instabilities. The imaginary component can still introduce oscillations, however the larger $\lambda_i h$, the smaller the imaginary part of $\alpha_{PF}$. We visualize this case in Figure 4(c).

**The importance of the largest eigenvalue** $\lambda_0$. The largest eigenvalue $\lambda_0$ plays an important part in the PF. Since $h\lambda_0 \geq h\lambda_i \quad \forall i$, $\lambda_0$ determines where in the above cases the PF is situated and thus whether there

are oscillations and unstable behavior in training. For all flows of the form we consider we can write:

$$\frac{dE(\boldsymbol{\theta})}{dt} = \frac{dE(\boldsymbol{\theta})}{d\boldsymbol{\theta}}^T \frac{d\boldsymbol{\theta}}{dt} = \nabla_{\boldsymbol{\theta}} E^T \sum_{i=0}^{D-1} \alpha(h\lambda_i) \nabla_{\boldsymbol{\theta}} E^T \mathbf{u}_i \mathbf{u}_i = \sum_{i=0}^{D-1} \alpha(h\lambda_i)(\nabla_{\boldsymbol{\theta}} E^T \mathbf{u}_i)^2 \tag{15}$$

and thus if $\alpha(h\lambda_i) \in \mathbb{R}$ and $\alpha(h\lambda_i) < 0 \ \forall i$ then $\frac{dE(\boldsymbol{\theta})}{dt} \le 0$ and following the corresponding flow minimises $E$. In the case of the PF this gets determined by $\lambda_0$. If $\lambda_0 < \frac{1}{h}$ then $\alpha_{PF}(h\lambda_i) < 0 \ \forall i$ (real stable case above) and the PF minimises E. If $1/h < \lambda_0 < \frac{2}{h}$ then $Re[\alpha_{PF}(h\lambda_i)] < 0 \ \forall i$ (complex stable case above) close to a gradient descent iteration $\lambda_i, \mathbf{u}_i \in \mathbb{R}$ we can write that $\frac{dRe[E(\boldsymbol{\theta})]}{dt} = \sum_{i=0}^{D-1} Re[\alpha_{PF}(h\lambda_i)](\nabla_{\boldsymbol{\theta}} E^T \mathbf{u}_i)^2$ and thus the real part of the loss function decreases. If $\lambda_0 > \frac{2}{h}$ then $Re[\alpha_{PF}(h\lambda_0)] > 0$ (unstable complex case above) and if $(\nabla_{\boldsymbol{\theta}} E^T \mathbf{u}_0)^2$ is sufficiently large we can no longer ascertain the behavior of $E$. We present a discrete time argument for this observation in Section A.8.1.

**Building intuition**. For quadratic objective $E(\boldsymbol{\theta}) = \frac{1}{2}\boldsymbol{\theta}^T A \boldsymbol{\theta}$ the PF describes gradient descent exactly. We show examples Figures 2 and 5. Unlike the NGF or the IGR flow, the PF captures the oscillatory and divergent behavior of gradient decent. Importantly, to capture the unstable behavior which occurs when $\lambda_0 > 1/h$ the imaginary part of the PF is needed. To expand intuition outside the quadratic case, we show the PF for the banana function (Rosenbrock, 1960) in Figure 6 and an additional example in 1D with a non-quadratic function (Figure 30 in the Appendix). In this case, the PF no longer follows the gradient descent trajectory exactly, but we still observe the importance of the PF in capturing instabilities of gradient descent; we also observe that adding non-principal terms can restabilize the trajectory.

**Remark 3.3** *For the banana function, the principal terms have a destabilizing effect when $h > 2/\lambda_0$ while the non principal terms can have a stabilizing effect.*

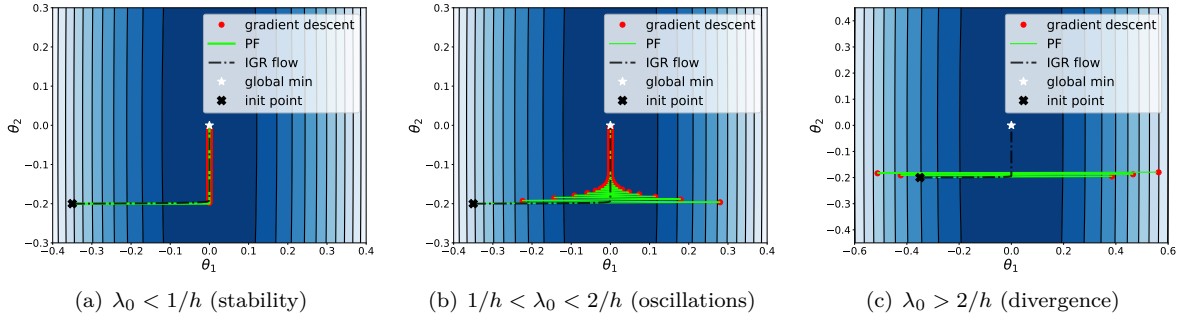

(a) $\lambda_0 < 1/h$ (stability)   (b) $1/h < \lambda_0 < 2/h$ (oscillations)   (c) $\lambda_0 > 2/h$ (divergence)

Figure 5: **Quadratic losses in 2 dimensions**. The PF captures the behavior of gradient descent exactly for quadratic losses, including oscillatory behavior and divergence.

### 3.2 The stability analysis of the principal flow

We now perform stability analysis on the PF, to understand how it can be used to predict certain behaviors of gradient descent around critical points of the loss function $E$. Consider $\boldsymbol{\theta}^*$ such a critical point, i.e $\nabla_{\boldsymbol{\theta}} E(\boldsymbol{\theta}^*) = \mathbf{0}$. For a critical point $\boldsymbol{\theta}^*$ to be exponentially asymptotically attractive, all eigenvalues of the Jacobian evaluated at $\boldsymbol{\theta}^*$ need to have strictly negative real part.

The PF has the following Jacobian at critical points (proof in Section A.4 in the Appendix):

$$J_{PF}(\boldsymbol{\theta}^*) = \sum_{i=0}^{D-1} \frac{\log(1 - h\lambda_i^*)}{h} \mathbf{u}_i^* \mathbf{u}_i^{*T} \tag{16}$$

where $\lambda_i^*$, $\mathbf{u}_i^*$ are the eigenvalues and eigenvectors of the Hessian $\nabla_{\boldsymbol{\theta}}^2 E(\boldsymbol{\theta}^*)$. We thus have that the eigenvalues of the Jacobian $J_{PF}(\boldsymbol{\theta}^*)$ at the critical point $\boldsymbol{\theta}^*$ are $\frac{1}{h} \log(1 - h\lambda_i^*)$ for $i = 1, \dots, D$.

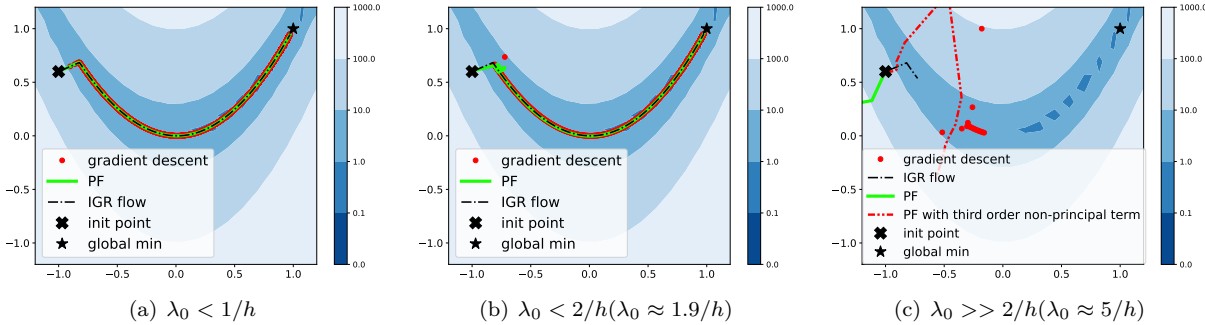

Figure 6: **Banana function**. The PF can capture instability and the gradient descent trajectory over many iterations when $\lambda_0$ is close to $2/h$. When $\lambda_0 >> 2/h$ (right) the PF does not track the GD trajectory over many gradient descent steps, but when including a non-principal term the flow is able to capture the general trajectory of gradient descent and unstable behavior of gradient descent.

**Local minima**. Suppose that $\boldsymbol{\theta}^*$ is a local minimum. Then all Hessian eigenvalues are non-negative $\lambda_i^* \geq 0$. We perform the stability analysis in cases given by the value of $\lambda_i^*$, corresponding to the cases in Section 3.1:

$h < 1/\lambda_i^*$. The corresponding eigenvalue of the Jacobian $\frac{1}{h}\log(1 - h\lambda_i^*)$ is negative, since $0 < 1 - h\lambda_i^* < 1$. The principal vector field is attractive in the corresponding eigenvector direction.

$h \in [1/\lambda_i^*, 2/\lambda_i^*)$. The corresponding eigenvalue of the Jacobian $\frac{1}{h}\log(1 - h\lambda_i^*) = \frac{1}{h}\log(h\lambda_i^* - 1) + i\frac{\pi}{h}$ is complex, with negative real part since since $h\lambda_i^* - 1 < 1$. The principal vector field is attractive in the corresponding eigenvector direction.

$h \geq 2/\lambda_i^*$. The corresponding eigenvalue of the Jacobian $\frac{1}{h}\log(1 - h\lambda_i^*) = \frac{1}{h}\log(h\lambda_i^* - 1) + i\frac{\pi}{h}$ is complex, with non-negative real part, since since $h\lambda_i^* - 1 \geq 1$. The principal vector field is not attractive in the corresponding eigenvector direction, and if $h > 2/\lambda_i^*$ it is repelled in the corresponding eigenvector direction.

The last case tells us that the PF is not always attracted to local minima, as it is not attractive in eigendrections where $h \geq 2/\lambda_i^*$. Thus **like gradient descent, the PF can be repelled around local minima for large learning rates**. This is in contrast to the NGF and the IGR flow, which always predict convergence around a local minimum: the eigenvalues of the NGF Jacobian are $-\lambda_i^*$, and for the IGR flow the eigenvalues are $-\lambda_i^* - \frac{h^2}{2}\lambda_i^{*2}$, both are negative when $\lambda_i^*$ is positive. For derivations see Section A.4.1 in the Appendix.

**Remark 3.4** *For quadratic losses, where the PF is exact, the results above recover the classical gradient descent result for quadratic losses namely that gradient descent convergences if $\lambda_0 < 2/h$, otherwise diverges.*

**Saddle points**. Suppose that $\boldsymbol{\theta}^*$ is a strict saddle point. In this case there exists $\lambda_s^*$ such that $\lambda_s^* < 0$. We want to analyse the behavior of the PF in the direction of the corresponding eigenvector $\mathbf{u}_s^*$. In that case, $\log(1 - h\lambda_s^*) > 0$ which entails that the PF is repelled in the eigendirections of strict saddle points. Note that this is also the case for the NGF since the corresponding eigenvalues of the Jacobian of the NGF would be $-\lambda_s^*$, also positive. Unlike the NGF however, the subspace of eigendirections that the PF is repelled by can be larger since it includes also eigendirections where $\lambda_i^* > 2/h > 0$.

## 4 Predicting neural network gradient descent dynamics with the principal flow

Computing the PF on large neural networks during training is computationally prohibitive, as it requires finding all eigenvalues of the Hessian matrix once for each step of the flow simulation, corresponding to many eigen-decompositions per gradient descent step. To build intuition about the PF for neural networks, we start with a small MLP for a 2 dimensional input regression problem, with random inputs and labels. Here we can understand the behavior of the PF since we can compute its modified vector field exactly and compare it with the behavior of gradient descent. We show results in Figure 7, where we visualize the norm of the difference between gradient descent parameters at each iteration and the parameters produced by the continuous time flows we compare with. We observe that *short term the principal flow is better than all other flows at tracking the behavior of gradient descent*. As the number of iterations increases however,

the PF accumulates error in the case of $\lambda_0 > 2/h$; this is likely due to the fact that while gradient descent parameters are real, this is not the case for the PF, as discussed in Remark 4.1. Since we are primarily concerned with using the PF to understand gradient descent for a small number of iterations this will be less of a concern in our experimental settings. Additional results which confirm the PF is better than the other flows at tracking gradient descent on a bigger network trained the UCI breast cancer dataset (Asuncion and Newman, 2007) are shown in Figure 32 in the Appendix.

**Remark 4.1** *On the multiple iteration behavior of the PF. We note that while gradient descent parameters are real for any iteration $\boldsymbol{\theta}_t$, $\boldsymbol{\theta}_{t+1}$, ... $\boldsymbol{\theta}_{t+n}$ when we approximate the behavior of gradient descent by initializing $\boldsymbol{\theta}(0) = \boldsymbol{\theta}_t$ and running the PF for time $nh$, there is nothing enforcing that $\boldsymbol{\theta}(h)$, ... $\boldsymbol{\theta}(nh)$ will be real when the PF is complex valued ($\lambda_0 > 1/h$). We also note that in that case the symmetric Hessian is not Hermitian and the eigenvalues and eigenvector of the Hessian will not be real; furthermore, the eigenvectors need not form a basis[1]. For long term trajectories (larger $n$), this can have an effect on long term error between gradient descent and PF trajectories, through an accumulating effect of the imaginary part in the PF. This can be mitigated by using the PF to understand the short term behavior of gradient descent (small $n$).*

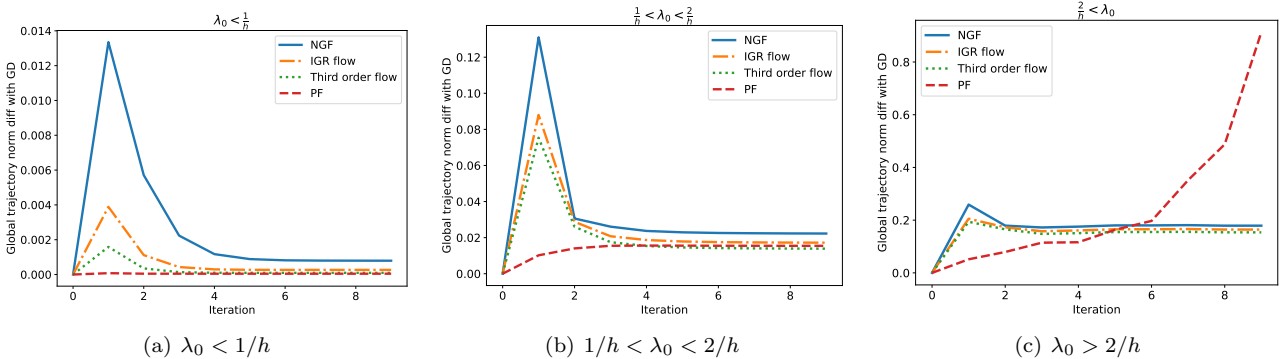

(a) $\lambda_0 < 1/h$        (b) $1/h < \lambda_0 < 2/h$        (c) $\lambda_0 > 2/h$

Figure 7: Error between gradient descent parameters and parameters obtained following continuous time flows for multiple iterations: $\|\boldsymbol{\theta}_n - \boldsymbol{\theta}(nh)\|$ with $\boldsymbol{\theta}(0) = \boldsymbol{\theta}_0$. For small $n$, the PF is better at capturing the behavior of gradient descent across all cases.

## 4.1 Predicting $\nabla_{\boldsymbol{\theta}} E^T \mathbf{u}_0$ using the principal flow

For large neural networks, instead of simulating the PF describing how the entire parameter vector changes in time we can use the PF to approximate changes in a scalar quantity only. This will allow us to compare the predictions of the PF against the predictions of the NGF and IGR flow on realistic settings. To do so, we first have to compute how the gradient changes in time:

**Corollary 4.1** *If $\boldsymbol{\theta}$ follows the PF, then: $(\dot{\nabla_{\boldsymbol{\theta}} E}) = \sum_{i=0}^{D-1} \frac{\log(1-h\lambda_i)}{h}(\nabla_{\boldsymbol{\theta}} E^T \mathbf{u}_i)\mathbf{u}_i$.*

This follows from applying the chain rule and using the definition of the PF. We contrast this with how the gradient evolves if the parameters follow the NGF:

**Corollary 4.2** *If $\boldsymbol{\theta}$ follows the NGF, then: $(\dot{\nabla_{\boldsymbol{\theta}} E}) = \sum_{i=0}^{D-1} -\lambda_i(\nabla_{\boldsymbol{\theta}} E^T \mathbf{u}_i)\mathbf{u}_i$*

**Corollary 4.3** *If $\boldsymbol{\theta}$ follows the IGR flow, then: $(\dot{\nabla_{\boldsymbol{\theta}} E}) = \sum_{i=0}^{D-1} - \left(\lambda_i + \frac{h}{2}\lambda_i^2\right)(\nabla_{\boldsymbol{\theta}} E^T \mathbf{u}_i)\mathbf{u}_i$*

We would like to use the above to assess how $\nabla_{\boldsymbol{\theta}} E^T \mathbf{u}_i$ changes in time under the above flows and check their predictions empirically against results obtained when training neural networks with gradient descent. Since

---

[1]To avoid the concern around the eigenvectors of the Hessian no longer forming a basis, one can use the Jordan normal form instead, as we show in Section A.7. We don't take this approach here as most of our following analysis is not affected, and is concerned with the behaviour of the PF around one gradient descent iteration. Furthermore, support of the Jordan normal form in code libraries is limited (especially for complex matrices), and we did not find this to be a significant issue in the experiments where we simulate the PF outside the quadratic case for a few iterations. We note, however, that mathematical analysis of long-term PF trajectories for general functions should use the Jordan normal form.

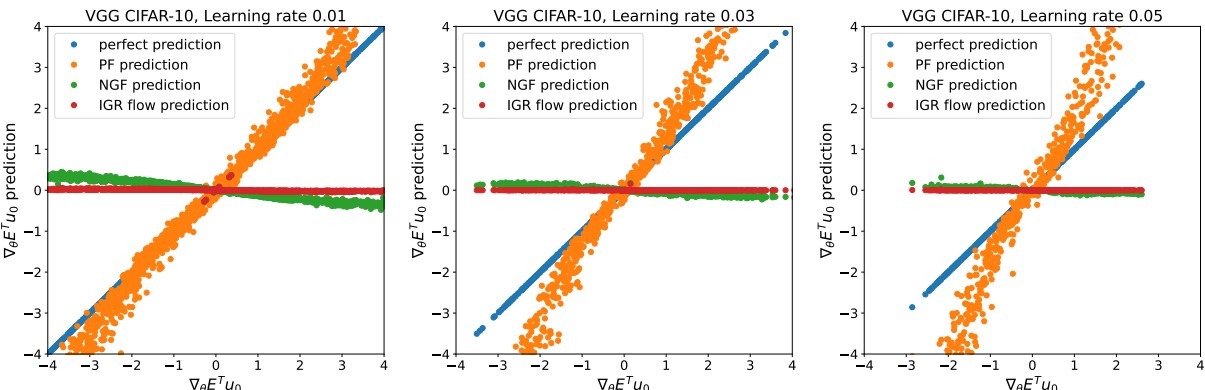

Figure 8: Predictions of $\nabla_{\boldsymbol{\theta}} E^T \mathbf{u}_0$ according to the NGF, IGR flow and the PF. On the $x$ axis we plot the value of $\nabla_{\boldsymbol{\theta}} E^T \mathbf{u}_0$ as measured empirically in training, and on the $y$ axis we plot the corresponding prediction according to the flows from the value of the dot product at the previous iteration. The 'exact match' line indicates a perfect prediction, the upper bound of performance. The PF performs best from all the compared flows, however for higher learning rates its performance degrades when $\nabla_{\boldsymbol{\theta}} E^T \mathbf{u}_0$ is large; this is due to the fact that the higher the learning rate and the higher the gradient norm, the more likely it is that the additional assumption we used that $\lambda_i, \mathbf{u}_i$ do not change does not hold.

$\mathbf{u}_i$ is an eigenvector of the Hessian it also changes in time according to the changes given by the corresponding flow, making $(\nabla_{\boldsymbol{\theta}} \dot{E}^T \mathbf{u}_i)$ difficult to calculate. Even when if we wrote an exact flow for $(\nabla_{\boldsymbol{\theta}} \dot{E}^T \mathbf{u}_i)$, it would be computationally challenging to simulate it since finding the new values of $\mathbf{u}_i$ would depend on the full Hessian and would lead to the same computational issues we are trying to avoid in the case of large neural networks. In order to mitigate these concerns, we will make the additional approximation that $\lambda_i$ and $\mathbf{u}_i$ do not change inside an iteration which will allow us to approximate changes to $\nabla_{\boldsymbol{\theta}} E^T \mathbf{u}_i$ and compare them against empirical observations. We note that we will not use this approximation for any other results.

**Remark 4.2** *If we assume that $\lambda_i, \mathbf{u}_i$ do not change between iterations, if $\boldsymbol{\theta}$ follows the PF then $(\nabla_{\boldsymbol{\theta}} \dot{E}^T \mathbf{u}_i) = \frac{\log(1-h\lambda_i)}{h} \nabla_{\boldsymbol{\theta}} E^T \mathbf{u}_i$.*

**Remark 4.3** *If we assume that $\lambda_i, \mathbf{u}_i$ do not change between iterations, if $\boldsymbol{\theta}$ follows the NGF we can write $(\nabla_{\boldsymbol{\theta}} \dot{E}^T \mathbf{u}_i) = -\lambda_i \nabla_{\boldsymbol{\theta}} E^T \mathbf{u}_i$.*

**Remark 4.4** *If we assume that $\lambda_i, \mathbf{u}_i$ do not change between iterations, if $\boldsymbol{\theta}$ follows the IGR flow we can write $(\nabla_{\boldsymbol{\theta}} \dot{E}^T \mathbf{u}_i) = -\left(\lambda_i + \frac{h}{2}\lambda_i^2\right) \nabla_{\boldsymbol{\theta}} E^T \mathbf{u}_i$.*

The above flows have the form $\dot{x} = cx$, with solution $x(t) = x(0)e^{ct}$. We can thus test these solutions empirically by training neural networks with gradient descent with learning rate $h$ and at each step compute $\nabla_{\boldsymbol{\theta}} E(\boldsymbol{\theta}_t)^T (\mathbf{u}_i)_{t-1}$ and compare it with the prediction $x(h)$ obtained from the solution from each flow initialized at the previous iteration, i.e. $x(0) = \nabla_{\boldsymbol{\theta}} E(\boldsymbol{\theta}_{t-1})^T (\mathbf{u}_i)_{t-1}$. We show results with a VGG model trained on CIFAR-10 in Figure 8. The results show that the PF is substantially better than the NGF and IGR flow at predicting the behavior of $\nabla_{\boldsymbol{\theta}} E^T \mathbf{u}_0$. Since the NGF and the IGR flow solutions scale the initial value by the inverse of an exponential of magnitude given by $\lambda_0$ for large $\lambda_0$ this leads to a small prediction, which is not aligned with what is observed empirically. We also note that the higher the value of $\nabla_{\boldsymbol{\theta}} E^T \mathbf{u}_0$, the worse the prediction of the PF; these are the areas where the approximations made in the above remarks are likely not to hold due to large gradient norms.

## 4.2 Around critical points: escaping sharp local minima and saddles

The stability analysis we performed in Section 3.2 showed the PF is repelled by local minima where $\lambda_0^* > 2/h$: that is, even if the model is close to a sharp local minima (with $\lambda_0^* > 2/h$), that local minima will not be attractive and training will continue until a shallow minima is reached. We provide experimental evidence to support that hypothesis in the context of neural networks in Figure 33 in the Appendix; these results are consistent with observations in the deep learning literature (Jastrzębski et al., 2018; Cohen et al., 2021).

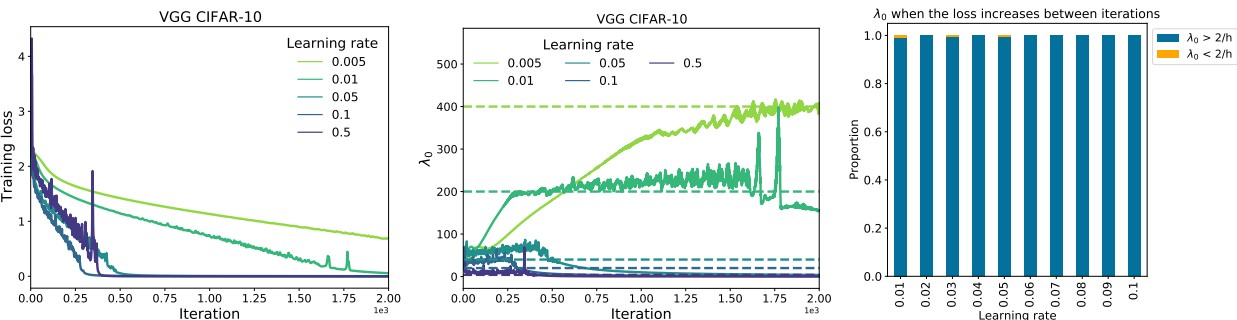

Figure 9: Edge of stability in neural networks (Cohen et al., 2021): instability occurs when $\lambda_0 > 2/h$.

Furthermore, while saddle points have long been considered a challenge with high dimensional optimisation (Dauphin et al., 2014) in practice gradient descent has not been observed to converge to saddles (Lee et al., 2016). Our analysis suggests that saddles will be repelled not only in the direction of strictly negative eigenvalues, but also in the eigendirections with large positive eigenvalues when large learning rates are used; this can explain why neural networks do not converge to non-strict saddles which exist in deep neural landscapes (Kawaguchi, 2016) but need not be repelling for the NGF and IGR flow (existing analyses of escaping saddle points by gradient descent apply only to strict saddles (Du et al., 2017; Lee et al., 2016)).

## 5   The principal flow, stability coefficients and edge of stability results

**Edge of stability results.** Cohen et al. (2021) did a thorough empirical study to show that when training deep neural networks with full batch gradient descent the largest eigenvalue of the Hessian, $\lambda_0$, keeps growing until reaching approximately $2/h$ (a phase of training they call *progressive sharpening*), after which it remains in that area; for mean squared losses this continues indefinitely while for cross entropy losses they show it further decreases later in training. They also show that instabilities in training occur when $\lambda_0 > 2/h$. Their empirical study spans neural architectures, data modalities and loss functions. We visualize the edge of stability behavior they observe in Figure 9; since we use a cross entropy loss $\lambda_0$ decreases later in training. We also visualize that iterations where the loss increases compared to the previous iteration overwhelmingly occur when $\lambda_0 > 2/h$. Cohen et al. (2021) also empirically observe that $\boldsymbol{\theta}^T \mathbf{u}_0$ has oscillatory behavior in the edge of stability area but is 0 or small outside it.

**Continuous-time models of gradient descent at edge of stability.** To investigate if existing continuous time flows and the PF capture gradient descent behavior at the edge of stability we train a 5 layer MLP on the toy UCI Iris dataset (Asuncion and Newman, 2007); this simple setting allows for the computation of the full eigenspectrum of the Hessian. We show results in Figure 10: the NGF and IGR flow have a larger error compared to the PF when predicting the parameters at the next gradient descent iteration in the edge of stability regime; the NGF and IGR flow predict the loss will decrease, while the PF captures the loss increase observed when following gradient descent. As we remarked in Section 2, the NGF and the IGR flow do not capture instabilities when the eigenvalues of the Hessian are positive, which has been remarked to be largely the case for neural network training through empirical studies (Sagun et al., 2017; Ghorbani et al., 2019; Papyan, 2018) and we observe here (Figure 34 in the Appendix). We spend the rest of the section using the PF to understand and model edge of stability phenomena using a continuous time approach.

**Connection with the principal flow: stability coefficients.** The PF captures the key quantities observed in the edge of stability phenomenon: the eigenvalues of the Hessian $\lambda_i$ and the threshold $2/h$. These quantities appear in the PF via the stability coefficient $sc_i = \frac{\log(1-h\lambda_i)}{h\lambda_i}\nabla_{\boldsymbol{\theta}}E^T\mathbf{u}_i = \alpha_{PF}(\lambda_i h)\nabla_{\boldsymbol{\theta}}E^T\mathbf{u}_i$ of eigendirection $\mathbf{u}_i$. Through the PF, by connecting the case analysis in Section 3.1 with existing and new empirical observations, we can shed light on the edge of stability behavior in deep learning.

*First phase of training (progressive sharpening):* $\lambda_0 < 2/h$. This entails $Re[sc_i] = Re[\alpha_{PF}(h\lambda_i)] \leq 0, \forall i$ (Real stable and complex stable cases of the analysis in Section 3.1). $\text{sign}(\alpha_{NGF}) = \text{sign}(\alpha_{PF}) = -1$ and following the PF minimises $E$ or its real part (Eq 15). To understand the behavior of $\lambda_0$, we now have to make use of empirical observations about the behavior of the NGF early in the training of neural networks. It has been

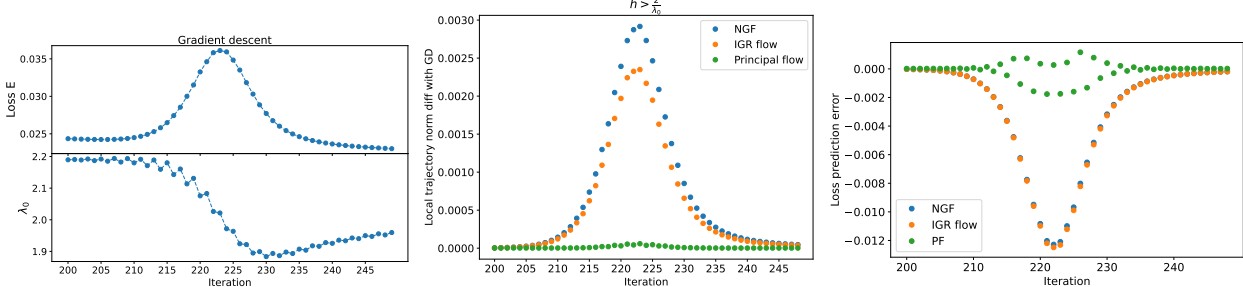

Figure 10: Comparing different continuous time models of gradient descent at the edge of stability area on a small 5 layer MLP, with 10 units per layer. We show the local parameter prediction error $\|\boldsymbol{\theta}_t - \boldsymbol{\theta}(h; \boldsymbol{\theta}_{t-1})\|$ for the NGF, IGR and PF flows (middle), as well as $E(\boldsymbol{\theta}(h; \boldsymbol{\theta}_{t-1})) - E(\boldsymbol{\theta}_t)$ (right).

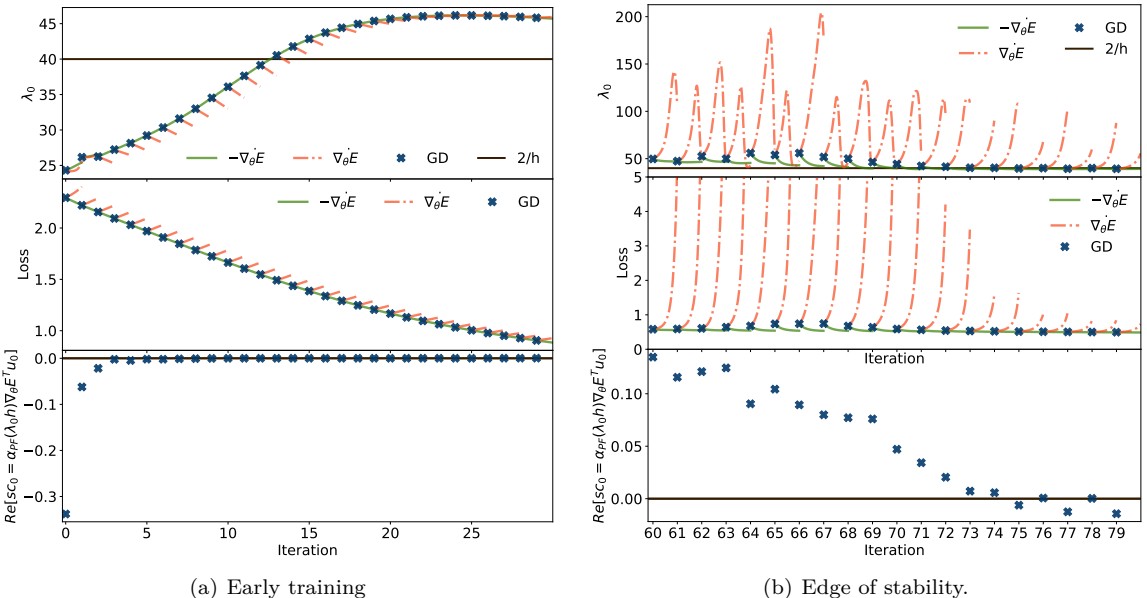

(a) Early training

(b) Edge of stability.

Figure 11: Understanding the edge of stability results using the PF on a 4 layer MLP: we plot the behavior of the NGF $\dot{\boldsymbol{\theta}} = -\nabla_{\boldsymbol{\theta}} E$ and the positive gradient flow $\dot{\boldsymbol{\theta}} = \nabla_{\boldsymbol{\theta}} E$ initialized at each gradient descent iteration parameters, and see that the behavior of gradient descent is connected to the behavior of the respective flow through the stability coefficient. *Figure 11(a) shows that even when $\lambda_0 > 2/h$, if the real part of the stability coefficient $sc_0$ is negative or close to 0, there are no instabilities in the loss and the eigenvalue $\lambda_0$ keeps increasing, as it does when following the NGF in that region.*

empirically observed that in early areas of training, $\lambda_0$ increases here when following the NGF (Cohen et al., 2021); we further show this in Figure 55 in the Appendix. Since in this part of training gradient descent follows closely the NGF, it exhibits similar behavior and $\lambda_0$ increases. We show this case in Figure 11(a).

*Second phase of training (edge of stability)* $\lambda_0 \geq 2/h$. This entails $Re[sc_0(\boldsymbol{\theta})] = Re[\alpha_{PF}(h\lambda_i)] \geq 0$. (Unstable complex case of the analysis in Section 3.1). We can no longer say that following the PF minimizes E. $\text{sign}(\alpha_{NGF}(h\lambda_0)) \neq \text{sign}(Re[(\alpha_{PF}(h\lambda_0)])$, since $\alpha_{NGF}(h\lambda_0) = -1$ and $\text{sign}(Re[(\alpha_{PF}(h\lambda_0)]) > 0$ meaning that in that direction gradient descent resembles the positive gradient flow $\dot{\boldsymbol{\theta}} = \nabla_{\boldsymbol{\theta}} E$ rather than the NGF. The positive gradient flow component can cause instabilities, and the strength of the instabilities depends on the stability coefficient $sc_0 = \alpha_{PF}(h\lambda_0)\nabla_{\boldsymbol{\theta}} E^T \mathbf{u}_0$. We show in Figures 11(b) and 13 how the behavior of the loss and $\lambda_0$ are affected by the behavior of the positive gradient flow when $\lambda_0 > 2/h$.

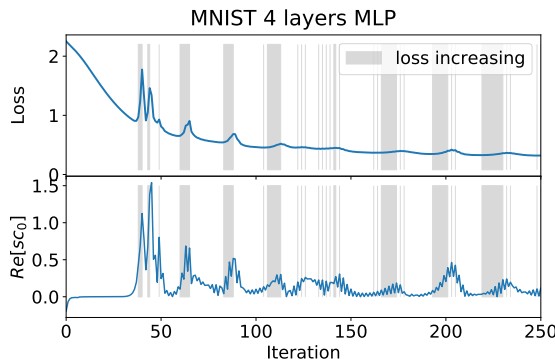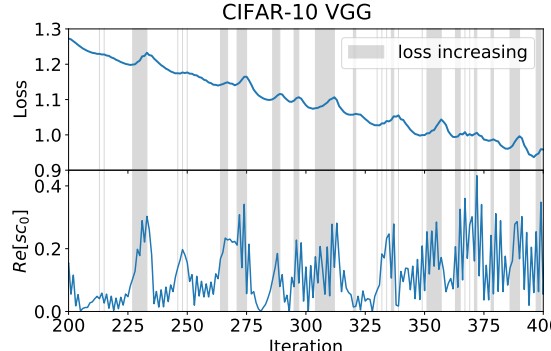

Figure 12: The loss function and stability coefficients: areas where the loss increases correspond to areas where the $sc_0$ is large. The highlighted areas correspond to regions where the loss increases.

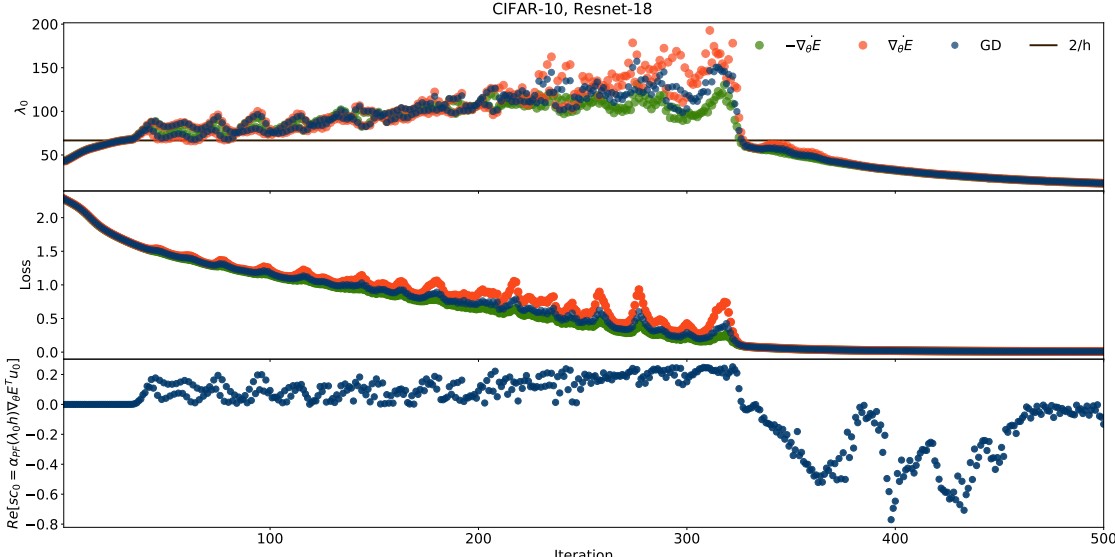

Figure 13: Loss instabilities, $\lambda_0$ and stabilitiy coefficients for CIFAR-10. Together with the behavior of gradient descent, we plot the behavior of the NGF and positive gradient flow initialized at $\boldsymbol{\theta}_t$ and simulated for time $h$ for each iteration $t$. The analysis we performed based on the PF suggests that when $Re[sc_0] > 0$ and large we should expected gradient descent to exhibit behaviors close to those of the positive gradient flow. What we observe empirically is that increases in loss value of gradient descent are proportional to the increase of the positive gradient flow in that area (can be seen best between iterations 200 and 350); the same behavior can be seen in relation to the eigenvalue $\lambda_0$.

**More than $\lambda_0$: the importance of stability coefficients**. While the sign of the real part of the stability coefficient $sc_0$ is determined by $\lambda_0$, its magnitude is modulated by the dot product $\nabla_{\boldsymbol{\theta}} E^T \mathbf{u}_0$, since $sc_0 = \alpha_{PF}(h\lambda_0)\nabla_{\boldsymbol{\theta}} E^T \mathbf{u}_0$. The magnitude of $\nabla_{\boldsymbol{\theta}} E^T \mathbf{u}_0$ plays an important role, since if $\lambda_0$ is the only eigenvalue greater than $2/h$ training is stable if $\nabla_{\boldsymbol{\theta}} E^T \mathbf{u}_0 = 0$, as we observe in Figure 11. *To understand instabilities, we have to look at stability coefficients, not only eigenvalues.* We show in Figure 12 how the instabilities in training can be related with the stability coefficient $sc_0$: the increases in loss occur when the corresponding $Re[sc_0]$ is positive and large. In Figure 13 we show results with the behavior of $\lambda_0$: $\lambda_0$ increases or decreases based on the behavior of the corresponding flow and the strength of the stability coefficient and that gets reflected in instabilities in the loss function; specifically when $\lambda_0 > 2/h$, we use the positive gradient flow and see how the strength of its fluctuations affect the changes both in the loss value and $\lambda_0$ of gradient descent. We show additional results in Figures 38 and 39 in the Appendix.

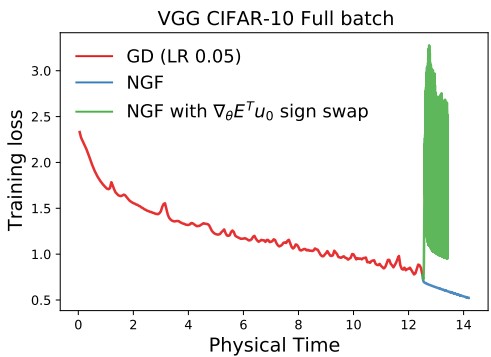 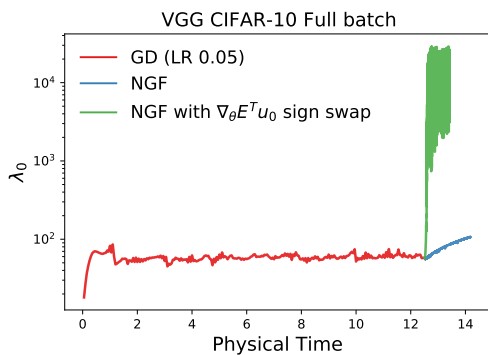

Figure 14: One eigendirection is sufficient to lead to instabilities. To create a situation similar to that of the PF, we construct a flow given by the NGF in all eigendirections but $\mathbf{u}_0$; in the direction of $\mathbf{u}_0$, we change the sign of the flow. This leads to the flow $\dot{\boldsymbol{\theta}} = \nabla_{\boldsymbol{\theta}} E^T \mathbf{u}_0 \mathbf{u}_0 + \sum_{i=1}^{D-1} -\nabla_{\boldsymbol{\theta}} E^T \mathbf{u}_i \mathbf{u}_i$. We show this flow can be very unstable when initialised in an edge of stability area.

**Is one eigendirection enough to cause instability?** One question that arises from the PF is whether the leading eigendirection $\mathbf{u}_0$ can be sufficient to cause instabilities, especially in the context of deep networks with millions of parameters. To assess this we train a model with gradient descent until it reaches the edge of stability ($\lambda_0 \approx 2/h$), after which we simulate the continuous flow $\dot{\boldsymbol{\theta}} = \nabla_{\boldsymbol{\theta}} E^T \mathbf{u}_0 \mathbf{u}_0 + \sum_{i=1}^{D-1} -\nabla_{\boldsymbol{\theta}} E^T \mathbf{u}_i \mathbf{u}_i$. The coefficients of the modified vector field of this flow are negative for all eigendirections except from $\mathbf{u}_0$, which is positive; this is also the case for the PF when $\lambda_0$ is the only eigenvalue greater than $2/h$. In Figure 14 we empirically show that a positive coefficient for $\mathbf{u}_0$ can be responsible for an increase in loss value and a significant change in $\lambda_0$ in neural network training.

**Decreasing the learning rate.** Cohen et al. (2021) show that if the edge of stability behavior is reached and the learning rate is decreased, the training stabilizes and $\lambda_0$ keeps increasing (Figure 35 in the Appendix). The PF tells us that decreasing the learning rate entails going from $Re[sc_0] \geq 0$ to $Re[sc_0] \leq 0$ since $\lambda_0 < 2/h$ after the learning rate change. Since all stability coefficients are now negative, this reduces instability. The increase in $\lambda_0$ is likely due to the behavior of the NGF in that area (as can be seen in Figure 14 when changing from gradient descent training to the NGF in an edge of stability area leads to an increase of $\lambda_0$).

**The behavior of $\nabla_{\boldsymbol{\theta}} E^T \mathbf{u}_0$.** The PF also allows us to explain the unstable behavior of $\nabla_{\boldsymbol{\theta}} E^T \mathbf{u}_0$ around edge of stability areas. As done in Section 4.1, we assume that $\lambda_i, \mathbf{u}_i$ do not change substantially between iterations and write $\nabla_{\boldsymbol{\theta}} \dot{E}^T \mathbf{u}_i = \frac{\log(1-h\lambda_i)}{h} \nabla_{\boldsymbol{\theta}} E^T \mathbf{u}_i$ under the PF, with solution $(\nabla_{\boldsymbol{\theta}} E^T \mathbf{u}_i)(t) = (\nabla_{\boldsymbol{\theta}} E^T \mathbf{u}_i)(0) e^{\frac{\log(1-h\lambda_i)}{h} t}$. This solution has different behavior depending on the value of $\lambda_0$ relative to $2/h$: decreasing below $2/h$ and increasing above $2/h$. We show this theoretically predicted behavior in Figure 15, alongside empirical behavior showcasing the fluctuation of $\nabla_{\boldsymbol{\theta}} E^T \mathbf{u}_0$ in the edge of stability area, which confirms the theoretical prediction. We also compute the prediction error of the proposed flow and show it can capture the dynamics of $\nabla_{\boldsymbol{\theta}} E^T \mathbf{u}_0$ closely in this setting. We present a discrete time argument for this observation in Section A.8.2. We note that the stable behavior early in training together with the oscillatory behavior of $\nabla_{\boldsymbol{\theta}} E^T \mathbf{u}_0$ in the edge of stability area which we predict and observe can explain the results of Cohen et al. (2021) on the behavior of $\boldsymbol{\theta}^T \mathbf{u}_0$, since $\boldsymbol{\theta}$ accumulates changes given by gradient updates.

**Why not more instability?** To determine why there isn't more instability in the edge of stability area we have to consider that neural networks are not quadratic, which has two effects. Firstly, when following the PF the landscape changes slightly locally; this leads to changes in stability coefficients and thus the behavior of gradient descent as we have consistently seen in the experiments in this section. Secondly, non-principal terms can have an effect; while we do not know all non-principal terms in Section B in the Appendix we provide a justification for why the non-principal term we do know (Eq 14) can have a stabilizing effect by inducing a regularisation pressure to minimise $\lambda_i(\nabla_{\boldsymbol{\theta}} E^T \mathbf{u}_i)^2$ in certain parts of the training landscape.

In this section we have shown the PF closely predicts the behavior of gradient descent in neural network training. This has led to additional insights, including the importance of stability coefficients in determining instabilities in gradient descent (Figures 11, 12, 13), causally showing one eigendirection is sufficient to cause

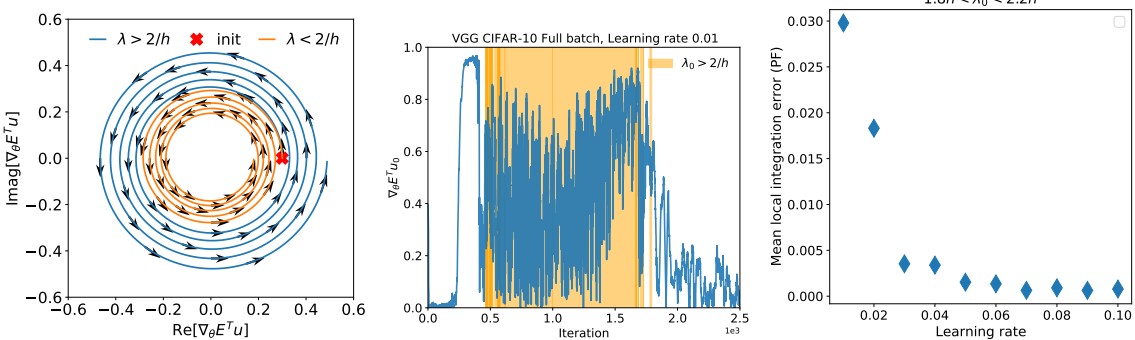

Figure 15: Predicting the unstable dynamics of $\nabla_{\boldsymbol{\theta}} E^T u$ in the edge of stability area ($\lambda \approx 2/h$) using the PF. Left: the predicted behavior of $\nabla_{\boldsymbol{\theta}} E^T u$ under $\nabla_{\boldsymbol{\theta}} \dot{E}^T \mathbf{u}_i = \frac{\log(1 - h\lambda_i)}{h} \nabla_{\boldsymbol{\theta}} E^T \mathbf{u}_i$, with an inflection point at $\lambda = 2/h$. Middle: empirical behavior of $\nabla_{\boldsymbol{\theta}} E^T u$ for a model shows instabilities in the edge of stability area (highlighted). Right: the approximation made to derive the flow is suitable around $\lambda \approx 2/h$.

instability (Figure 14) and change and being able to closely predict the behavior of the dot product between the gradient and the largest eigenvector (Figure 15). This evidence suggests that the PF captures significant aspects of the behavior of gradient descent in deep learning; this is likely due to the specific structure of neural network models. While we take a continuous time approach, a discrete time approach can be used to motivate some of our observations (Section A.8); this is complementary to our approach but nonetheless related, since it also does not account for higher order derivatives of the loss and further suggests the strength of a quadratic approximation of the loss in the case of neural networks, as observed by Cohen et al. (2021).

## 6    Stabilizing training by adjusting discretization drift

The PF allows us to understand not only how gradient descent differs from the trajectory given by the NGF, but also when they follow each other very closely. Understanding when gradient descent behaves like the NGF flow reveals when the existing analyses of gradient descent using the NGF discussed in Section 2 are valid. It also has practical implications, since in areas where gradient descent follows the NGF closely training can be sped up by increasing the learning rate. Prior works have empirically observed that gradient descent follows the NGF early in neural network training (Cohen et al., 2021) and this observation can be used to explain why decaying learning rates  (Loshchilov and Hutter, 2016) or learning rate warm up (He et al., 2019) are successful when training neural networks: having a high learning rate in areas where the drift is small will not cause instabilities and can speed up training while decaying the learning rate avoids instabilities later in training when the drift is larger.

### 6.1    $\nabla_{\boldsymbol{\theta}}^2 E \nabla_{\boldsymbol{\theta}} E$ determines discretization drift

In previous sections we have seen that the Hessian plays an important role in defining the PF and in training instabilities. We now want to quantify the difference between the NGF and the PF in order to understand when the NGF can be used as a model of gradient descent. We find that:

**Remark 6.1** *In a region of the space where $\nabla_{\boldsymbol{\theta}}^2 E \nabla_{\boldsymbol{\theta}} E = \mathbf{0}$ the PF is the same as the NGF.*

To see why, we can expand

$$\nabla_{\boldsymbol{\theta}}^2 E \nabla_{\boldsymbol{\theta}} E = \sum_{i=0}^{D-1} \lambda_i \nabla_{\boldsymbol{\theta}} E^T \mathbf{u}_i \mathbf{u}_i. \tag{17}$$

If $\nabla_{\boldsymbol{\theta}}^2 E \nabla_{\boldsymbol{\theta}} E = \mathbf{0}$ we have that $\lambda_j \nabla_{\boldsymbol{\theta}} E^T \mathbf{u}_j = 0, \forall j \in \{1, .., D\}$, thus either $\lambda_j = 0$ leading to $\alpha_{NGF}(h\lambda_j) = \alpha_{PF}(h\lambda_j) = -1$ or $\nabla_{\boldsymbol{\theta}} E^T \mathbf{u}_j = 0$. Then $\dot{\boldsymbol{\theta}} = \sum_{i=0}^{D-1} \alpha_{PF}(h\lambda_i)(\nabla_{\boldsymbol{\theta}} E^T \mathbf{u}_i) \mathbf{u}_i = \sum_{i=0}^{D-1} \alpha_{NGF}(h\lambda_i)(\nabla_{\boldsymbol{\theta}} E^T \mathbf{u}_i) \mathbf{u}_i$.

Thus comparing the PF with the NGF reveals an important quantity: $\nabla_{\boldsymbol{\theta}}^2 E \nabla_{\boldsymbol{\theta}} E$. Further investigating this quantity reveals it has a connection with the total drift, since:

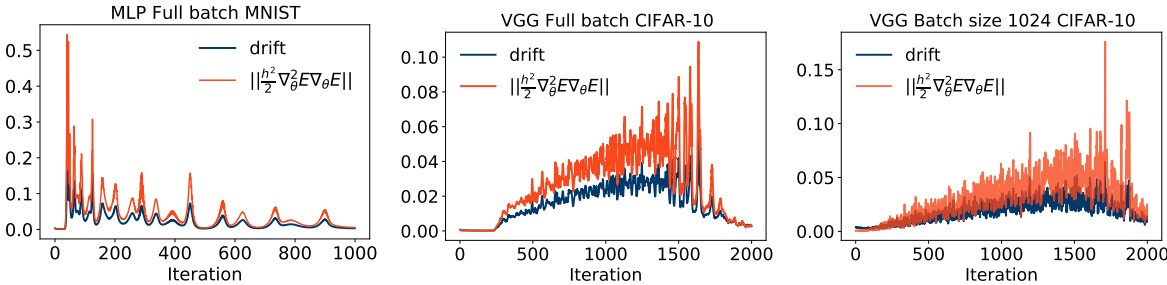

Figure 16: Connection between $||\nabla^2_{\boldsymbol{\theta}} E \nabla_{\boldsymbol{\theta}} E||$ and the per iteration drift as measured during training.

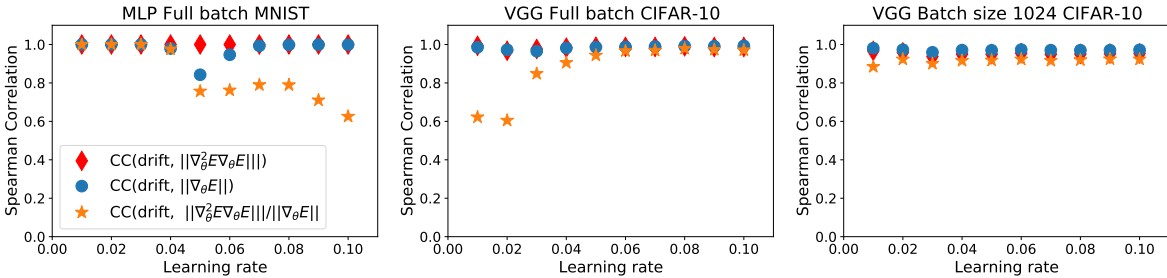

Figure 17: Correlation between $||\nabla^2_{\boldsymbol{\theta}} E \nabla_{\boldsymbol{\theta}} E||$ and the per iteration drift. Since $||\nabla^2_{\boldsymbol{\theta}} E \nabla_{\boldsymbol{\theta}} E|| = \left( ||\nabla^2_{\boldsymbol{\theta}} E \nabla_{\boldsymbol{\theta}} E|| \right) / ||\nabla_{\boldsymbol{\theta}} E|| ||\nabla_{\boldsymbol{\theta}} E||$, we plot the correlation with the individual terms as well.

**Theorem 6.1** *The discretization drift (error between gradient descent and the NGF) after 1 iteration $\boldsymbol{\theta}_t = \boldsymbol{\theta}_t - h\nabla_{\boldsymbol{\theta}} E(\boldsymbol{\theta}_{t-1})$ is $\frac{h^2}{2}\nabla^2_{\boldsymbol{\theta}} E(\boldsymbol{\theta}')\nabla_{\boldsymbol{\theta}} E(\boldsymbol{\theta}')$ for a set of parameters $\boldsymbol{\theta}'$ in the neighborhood of $\boldsymbol{\theta}_{t-1}$.*

This follows from the Taylor reminder theorem in mean value form (proof in Section A.10). This leads to:

**Corollary 6.1** *In a region of space where $\nabla^2_{\boldsymbol{\theta}} E \nabla_{\boldsymbol{\theta}} E = \mathbf{0}$ gradient descent follows the NGF.*

Thus the PF revealed $\nabla^2_{\boldsymbol{\theta}} E \nabla_{\boldsymbol{\theta}} E$ as a core quantity in the discretisation drift of gradient descent. To further see the connection between with the PF consider that $\left\| \nabla^2_{\boldsymbol{\theta}} E \nabla_{\boldsymbol{\theta}} E \right\|^2 = \left\| \sum_{i=0}^{D-1} \lambda_i \nabla_{\boldsymbol{\theta}} E^T \mathbf{u}_i \mathbf{u}_i \right\|^2 = \sum_{i=0}^{D-1} \left\| \lambda_i \nabla_{\boldsymbol{\theta}} E^T \mathbf{u}_i \right\|^2$; the higher each term in the sum, the higher the difference between the NGF and the PF. To measure the connection between per iteration drift and $\left\| \nabla^2_{\boldsymbol{\theta}} E \nabla_{\boldsymbol{\theta}} E \right\|$ in neural network training we approximate it via $\left\| \boldsymbol{\theta}_t - \widetilde{NGF}(\boldsymbol{\theta}_{t-1}, h) \right\|$ where $\widetilde{NGF}$ is the numerical approximation to the NGF initialised at $\boldsymbol{\theta}_{t-1}$. Results in Figures 16 and 17 show the strong correlation between per iteration drift and $\left\| \nabla^2_{\boldsymbol{\theta}} E \nabla_{\boldsymbol{\theta}} E \right\|$ throughout training and across learning rates. Since Theorem 6.1 tells us the form of the drift but not the exact value of $\boldsymbol{\theta}'$, we have used $\boldsymbol{\theta}_{t-1}$ instead to evaluate $\left\| \nabla^2_{\boldsymbol{\theta}} E \nabla_{\boldsymbol{\theta}} E \right\|$ and thus some error exists.

Understanding this connection is advantageous since computing discretization drift is computationally expensive as it requires simulating the continuous time NGF but computing $\left\| \nabla^2_{\boldsymbol{\theta}} E \nabla_{\boldsymbol{\theta}} E \right\|$ via Hessian-vector products is cheaper and approximations are available, such as $\nabla^2_{\boldsymbol{\theta}} E \nabla_{\boldsymbol{\theta}} E \approx \frac{\nabla_{\boldsymbol{\theta}} E(\boldsymbol{\theta} + \epsilon \nabla_{\boldsymbol{\theta}} E) - \nabla_{\boldsymbol{\theta}} E(\boldsymbol{\theta})}{\epsilon}$ which only requires an additional backward pass Geiping et al. (2021).

## 6.2 Drift adjusted learning rate (DAL)

A natural question to ask is how to use the correlation between $\left\| \nabla^2_{\boldsymbol{\theta}} E \nabla_{\boldsymbol{\theta}} E \right\|$ and the iteration drift to improve training stability; $\left\| \nabla^2_{\boldsymbol{\theta}} E \nabla_{\boldsymbol{\theta}} E \right\|$ captures all the quantities we have shown to be relevant to instability highlighted by the PF: $\lambda_i$ and $\nabla_{\boldsymbol{\theta}} E^T \mathbf{u}_i$ (Eq. 17). One way to use this information is to adapt the learning rate of the gradient descent update, such as using $\frac{2}{\left\| \nabla^2_{\boldsymbol{\theta}} E \nabla_{\boldsymbol{\theta}} E \right\|}$ as the learning rate. This learning rate slows down training when the drift is large — areas where instabilities are likely to occur — and it speeds up training in regions of low drift — areas where instabilities are unlikely to occur. Computing the norm of

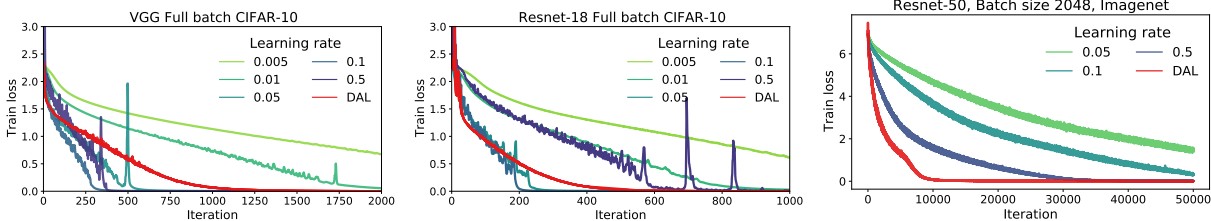

Figure 18: DAL: using the learning rate $\dfrac{2}{\left\|\nabla_{\boldsymbol{\theta}}^2 E \hat{\mathbf{g}}(\boldsymbol{\theta})\right\|}$ results in improved stability without requiring a hyper-parameter sweep.

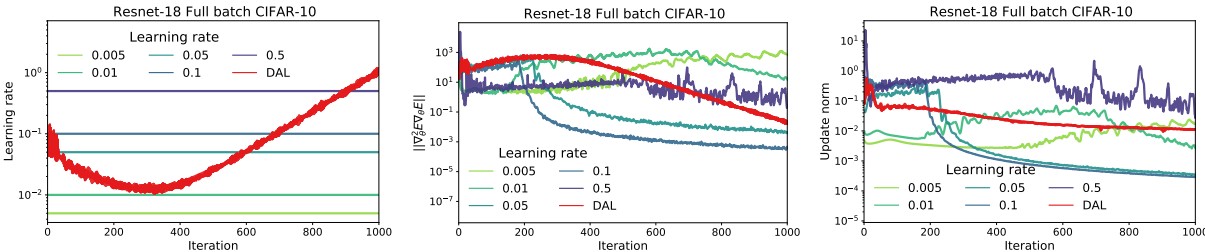

Figure 19: Key quantities in DAL versus fixed learning rate training: learning rate, and update norms.

the update provided by this learning rate shows a challenge however since $2/\left\|\nabla_{\boldsymbol{\theta}}^2 E \nabla_{\boldsymbol{\theta}} E\right\| \geq \frac{2}{\lambda_0 \|\nabla_{\boldsymbol{\theta}} E\|}$; this implies that when using this learning rate the norm of the gradient descent update will never be 0 and thus training will not result in convergence. Furthermore, the magnitude of the parameter update will be independent of the gradient norm. To reinstate the gradient norm, we propose using the learning rate

$$h(\boldsymbol{\theta}) = \frac{2}{\left\|\nabla_{\boldsymbol{\theta}}^2 E \nabla_{\boldsymbol{\theta}} E\right\| / \|\nabla_{\boldsymbol{\theta}} E\|} = \frac{2}{\left\|\nabla_{\boldsymbol{\theta}}^2 E \hat{\mathbf{g}}(\boldsymbol{\theta})\right\|} \tag{18}$$

where $\hat{\mathbf{g}}(\boldsymbol{\theta})$ is the unit normalised gradient $\nabla_{\boldsymbol{\theta}} E / \|\nabla_{\boldsymbol{\theta}} E\|$. We will call this learning rate **DAL** (Drift Adjusted Learning rate). As shown in Figure 16, $\left\|\nabla_{\boldsymbol{\theta}}^2 E \hat{\mathbf{g}}(\boldsymbol{\theta})\right\|$ has a strong correlation with the per iteration drift. Another interpretation of DAL can be provided through a signal to noise perspective: the size of the learning signal obtained by minimising $E$ is that of the update $h \|\nabla_{\boldsymbol{\theta}} E\|$, while the norm of the noise coming from the drift can be approximated as $\frac{h^2}{2} \left\|\nabla_{\boldsymbol{\theta}}^2 E \nabla_{\boldsymbol{\theta}} E\right\|$, thus the 'signal to noise ratio' can be approximated as $h \|\nabla_{\boldsymbol{\theta}} E\| / (\frac{h^2}{2} \left\|\nabla_{\boldsymbol{\theta}}^2 E \nabla_{\boldsymbol{\theta}} E\right\|) = 2/(h \left\|\nabla_{\boldsymbol{\theta}}^2 E \hat{\mathbf{g}}(\boldsymbol{\theta})\right\|)$, which when using DAL (Eq 18) is 1; thus DAL can be seen as balancing the gradient signal and the regularising drift noise in gradient descent training.

We use DAL to set the learning rate and show results across architectures, models and datasets in Figures 18 (with additional results in Figure 42 in the Appendix). *Despite not requiring a learning rate sweep, DAL is stable compared to using fixed learning rates.* To provide intuition about DAL, we show the learning rate and the update norm in Figure 19: for DAL the learning rate decreases in training after which it slowly increases when reaching areas with low drift. Compared to larger learning static learning rates where the update norm can increase in the edge of stability area with DAL the update norm steadily decreases in training.

### 6.3 The trade-off between stability and performance

Since we are interested in understanding the optimisation dynamics of gradient descent, we have so far focused on training performance. We now try to move our attention to test performance and generalization. Previous works (Li et al., 2019; Barrett and Dherin, 2021; Jastrzebski et al., 2019) have shown that higher learning rates lead to better generalization performance. We now try to further connect this information with the per iteration drift and the PF. To do so, we use learning rates with various degrees of sensitivity to iteration drift using DAL-$p$:

$$h_p(\boldsymbol{\theta}) = \frac{2}{\left(\left\|\nabla_{\boldsymbol{\theta}}^2 E \hat{\mathbf{g}}(\boldsymbol{\theta})\right\|\right)^p} \tag{19}$$

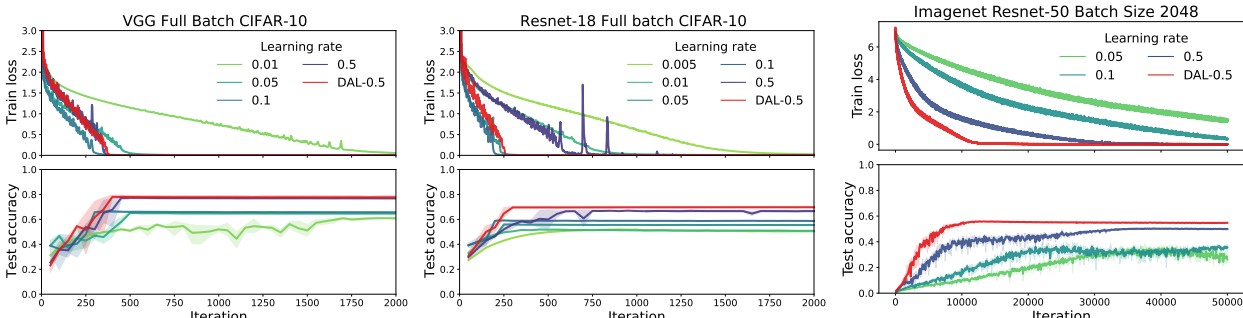

Figure 20: DAL-0.5: increased training speed and generalization compared to a sweep of fixed learning rates.

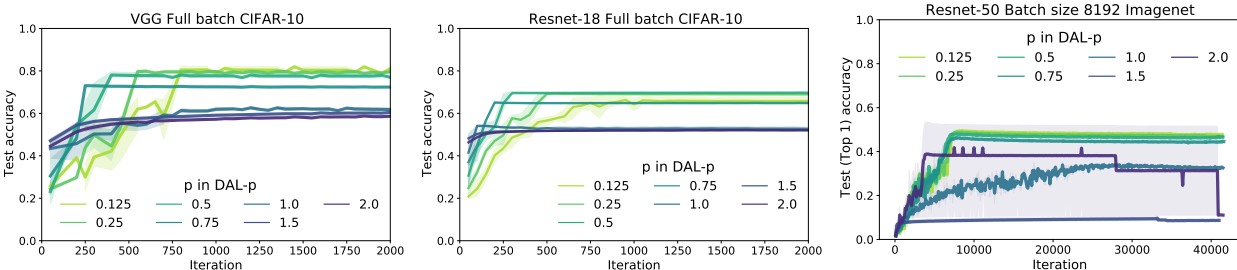

Figure 21: DAL-$p$ sweep: discretization drift helps test performance at the cost of stability. Corresponding training curves and loss functions are present in the Figure 44 in the Appendix; results showing the same trends across various batch sizes are shown in Figure 41.

The higher $p$, the slower the training and less drift there is; the lower $p$, there is more drift. We start with extensive experiments with $p = 0.5$, which we show in Figure 20, and show more results in Figure 43. Compared to $p = 1$ (DAL), there is faster training but at times also more instability. Performance on the test set shows that DAL-0.5 performs as well or better than when using fixed learning rates.

**Remark 6.2** *We find that across datasets and batch sizes, DAL-0.5 performs best in terms of the stability generalization trade-off and in these settings can be used as a drop in replacement for a learning rate sweep.*

To further investigate the connection between drift and test set performance, we perform a set of sweeps over the power $p$ and show results in Figure 21. These results show that the higher the drift (the smaller $p$), the more generalization; additional results across batch sizes showing the same trend are shown in Figure 41 in the Appendix. We also show in Figure 22 the correlation between mean per iteration drift and test accuracy both for learning rate and DAL-$p$ sweeps. The results consistently show that the higher the mean iteration drift, the higher the test accuracy. We also show that the mean iteration drift has a connection to the largest eigenvalue $\lambda_0$: the higher the drift, the smaller $\lambda_0$. These results add further evidence to the idea that discretization drift is beneficial for generalization performance in the deep learning setting. We also notice that DAL-$p$ with smaller values of $p$ leads to a small $\lambda_0$ compared to vanilla gradient descent even when large learning rates are used for the latter; this could explain its generalisation capabilities as lower sharpness has been connected to generalisation in previous works (Keskar et al., 2016; Jastrzębski et al., 2018; Foret et al., 2020). To consolidate these results, we use the method of Li et al. (2018) to visualise the loss landscape learned by DAL-$p$ compared to that learned using gradient descent, and observe that *even when reporting similar accuracies, DAL-p converges to a flatter landscape*; this is observed even when small batch sizes are used. Results are shown in Figures 46, 48, 50 in the Appendix.

Inspired by understanding when the PF is close to the NGF, in this section we investigated the total discretisation drift of gradient descent. This led us to DAL-$p$, a method to automatically set the learning rate based on approximation to the per iteration drift of gradient descent; we have seen that DAL produces stable training and further connected discretisation drift, generalisation and flat landscapes as measured by leading Hessian eigenvalues.

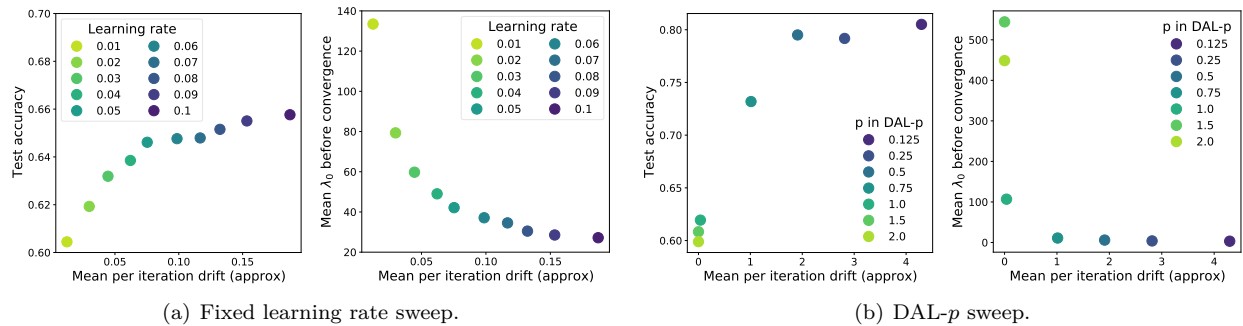

(a) Fixed learning rate sweep.        (b) DAL-$p$ sweep.

Figure 22: The correlation between drift, test set performance and $\lambda_0$ in full batch training on CIFAR-10. The same pattern can be seen in SGD results in Figure 53.

# 7    Future work

**Beyond gradient descent**. In this work we focused on understanding vanilla gradient descent. Understanding discretization drift via the PF can be beneficial for improving other gradient based optimization algorithms as well, as we briefly illustrate for momentum updates with decay $m$ and learning rate $h$:

$$\mathbf{v}_t = m\mathbf{v}_{t-1} - h\nabla_{\boldsymbol{\theta}}E(\boldsymbol{\theta}_{t-1}); \qquad\qquad \boldsymbol{\theta}_t = \boldsymbol{\theta}_{t-1} + \mathbf{v}_t \qquad\qquad (20)$$

We can scale $\nabla_{\boldsymbol{\theta}}E(\boldsymbol{\theta}_{t-1})$ in the above not by a fixed learning rate $h$, but by adjusting the learning rate according to the approximation to the drift. This has two advantages: it removes the need for a learning rate sweep and it uses local landscape information in adapting the moving average, such that in areas of large drift the contribution is decreased, while it is increased in areas where the drift is small (a more formal justification is provided in Section A.10). This leads to the following updates:

$$\mathbf{v}_t = m\mathbf{v}_{t-1} - \frac{1}{2||\nabla_{\boldsymbol{\theta}}^2 E(\boldsymbol{\theta}_{t-1})\hat{\mathbf{g}}(\boldsymbol{\theta})(\boldsymbol{\theta}_{t-1})||}\nabla_{\boldsymbol{\theta}}E(\boldsymbol{\theta}_{t-1}) \qquad\qquad \boldsymbol{\theta}_t = \boldsymbol{\theta}_{t-1} + \mathbf{v}_t \qquad (21)$$

As with DAL-$p$, we can use powers to control the stability performance trade-off: the lower $p$, the more the current update contribution is reduced in high drift (instability) areas. We tested this approach on Imagenet and show results in Figure 23. The results show that integrating drift information improves the speed of convergence compared to standard gradient descent (Figure 21), and leads to more stable training compared to using a fixed learning rate. We present additional experimental results in the Appendix.

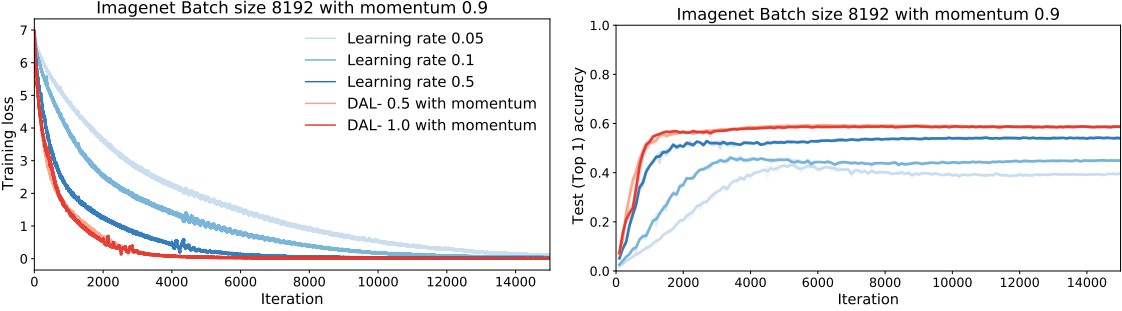

Figure 23: DAL with momentum: integrating drift information results in faster and more stable training compared to a fixed learning rate sweep. Compared to vanilla gradient descent there is also a significant performance and convergence speed boost.

Just as momentum is a common staple of optimization algorithms, so are adaptive schemes such as Adam (Kingma and Ba, 2015) and Adagrad (Duchi et al., 2011), which adjust the step taken for each

parameter independently. We can also use the knowledge from the PF to set a per parameter learning rate: instead of using $\left\| \nabla_{\boldsymbol{\theta}}^2 E(\boldsymbol{\theta}_{t-1}) \nabla_{\boldsymbol{\theta}} E(\boldsymbol{\theta}_{t-1}) \right\|$ to set a global learning rate, we can use the per parameter information provided by $\nabla_{\boldsymbol{\theta}}^2 E(\boldsymbol{\theta}_{t-1}) \nabla_{\boldsymbol{\theta}} E(\boldsymbol{\theta}_{t-1})$ to adapt the learning rate of each parameter. We present preliminary results in the Appendix (Figures 51 and 52). The above two approaches (momentum and per-parameter learning rate adaptation) can be combined, bringing us closer to the most commonly used deep learning optimization algorithms. While we do not explore this avenue here, we are hopeful that this understanding of discretization drift can be leveraged further to stabilize and improve deep learning optimization.

**Non-principal terms**. This work focuses on understanding the effects of the PF on the behavior of gradient descent. The principal terms however are not the only terms in the discretization drift: we have found one non-principal term (Eq 8) and have seen that it can have a stabilising effect (Figure 6). We provide a preliminary explanation for the stabilising effect of this non-principal term together with results measuring its value in neural network training in Section B in the Appendix. One promising avenue of non-principal terms is theoretically modelling the change of the eigenvalues $\lambda_i$ in time; another promising direction is that of implicit regularisation: while existing work which uses BEA in deep learning has found important implicit regularisation effects (Barrett and Dherin, 2021; Smith et al., 2021; Rosca et al., 2021), we have shown here that considering only effects of $\mathcal{O}(h^3)$ is not sufficient to capture the intricacies of gradient descent, which suggests that other implicit regularisation effects could be uncovered using the non-principal terms.

**Neural network theory**. Many theoretical works studying at gradient descent in the neural network context use the NGF (Du et al., 2018; Elkabetz and Cohen, 2021; Kunin et al., 2021; Jacot et al., 2018). We posit that replacing NGF in these theoretical contexts with PF may yield interesting results. In contrast to the NGF, the PF allows the incorporation of the learning rate into the analysis, and unlike existing continuous time models of gradient descent, it can model unstable behaviors observed in the discrete case. An example can be seen using the Neural Tangent Kernel: Jacot et al. (2018) model gradient descent using the NGF to show that in the infinite wide limit gradient descent for neural networks follows kernel gradient descent. The PF can be incorporated in this analysis either by replacing the NGF with the PF as a model of gradient descent or by studying the difference in the PF for infinitely wide and finite width networks, since discretisation drift could be responsible for the observed gap between finite and infinite networks in the large learning rate case (Lee et al., 2020).

## 8 Related work

**Modified flows for deep learning optimization**. Barrett and Dherin (2021) found the first order correction modified flow for gradient descent using BEA and uncovered its regularization effects; they were the first to show the power of BEA in the deep learning context. Smith et al. (2021) find the first order error correction term in expectation during one epoch of stochastic gradient descent. Modified flows have also been used for other optimizers than vanilla gradient descent: Franca et al. (2020); Shi et al. (2021) compare momentum and Nesterov accelerated momentum; Kunin et al. (2021) study the symmetries of deep neural networks and use modified vector fields to show commonly used discrete updates break conservation laws present when using the NGF (for gradient descent they use the IGR flow while for momentum and weight decay they introduce different flows); Kovachki and Stuart (2021) use modified flows to understand the behavior of momentum by approximating Hamiltonian systems; França et al. (2021) construct optimizers controlling their stability and convergence rates while Li et al. (2017) construct optimizers with adaptive learning rates in the context of stochastic differential equations. In the context of two-player games, Rosca et al. (2021) compute the first order BEA correction terms while Chavdarova et al. (2021) use high-resolution differential equations to shed light on the properties of different saddle point optimizers.

In concurrent work Miyagawa (2022) use BEA to find a modified flow coined 'Equations of Motion' (EOM) to describe gradient descent and find higher order terms, including non-principal terms; their focus is however on EOM(1), which is the IGR flow, which they use to understand scale and translation invariant layers. Their approach does not expand to complex space and does not capture the instabilities studied here (see also the discussion on the difference between the full modified flow provided by BEA and the PF in Section 3).

**Edge of stability and the importance of the Hessian**. There have been a number of empirical studies on the Hessian in gradient descent. Cohen et al. (2021) observed the edge of stability behavior and performed an extensive study which led to many empirical observations used in this work. Jastrzębski et al. (2018) performed a similar study in the context of stochastic gradient descent. Sagun et al. (2017); Ghorbani et al.

(2019); Papyan (2018) approximate the entire spectrum of the Hessian, and show that there are only a few negative eigenvalues, plenty of eigenvalues centered around 0, and a few positive eigenvalues with large magnitude. Similarly, Gur-Ari et al. (2018) discuss how gradient descent operates in a small subspace. Lewkowycz et al. (2020) discuss the large learning rate catapult in deep learning when the largest eigenvalue exceeds $2/h$. Gilmer et al. (2021) assess the effects of the largest Hessian eigenvalue in a large number of empirical settings.

There have been a series of concurrent works aimed at theoretically explaining the empirical results above. Ahn et al. (2022) connect the edge of stability behavior with what they coin as the 'relative progress ratio': $\frac{E(\boldsymbol{\theta}-h\nabla_{\boldsymbol{\theta}}E)-E(\boldsymbol{\theta})}{h\|\nabla_{\boldsymbol{\theta}}E\|^2}$, which they empirically show is 0 in stable areas of training and 1 in the edge of stability areas. To see the connection between the relative progress ratio and the quantities discussed in this paper, one can perform a Taylor expansion on $\frac{E(\boldsymbol{\theta}-h\nabla_{\boldsymbol{\theta}}E)-E(\boldsymbol{\theta})}{h\|\nabla_{\boldsymbol{\theta}}E\|^2} \approx \frac{-h\nabla_{\boldsymbol{\theta}}E^T\nabla_{\boldsymbol{\theta}}E+h^2/2\nabla_{\boldsymbol{\theta}}E^T\nabla_{\boldsymbol{\theta}}^2E\nabla_{\boldsymbol{\theta}}E}{h\|\nabla_{\boldsymbol{\theta}}E\|^2} = -1 + h/2\frac{\nabla_{\boldsymbol{\theta}}E^T\nabla_{\boldsymbol{\theta}}^2E\nabla_{\boldsymbol{\theta}}E}{\|\nabla_{\boldsymbol{\theta}}E\|^2}$. While this ratio is related to the quantities we discuss, we also note significant differences: it is a scalar, and not a parameter length vector and thus does not capture per eigendirection behavior as we see with the stability coefficients (Section 5). Arora et al. (2022) prove the edge of stability result occurs under certain conditions either on the learning rate or on the loss function. Ma et al. (2022) empirically observe the multi-scale structure of the loss landscape in neural networks and use it to theoretically explain the edge of stability behavior of gradient descent. Chen and Bruna (2022) use low dimensional theoretical insights around a local minima to understand the edge of stability behavior. Damian et al. (2022) use a cubic Taylor expansion to show that gradient descent follows the trajectory of a projected method which ensures that $\lambda_0 < 2/h$ and $\nabla_{\boldsymbol{\theta}}E^T\mathbf{u} = 0$; their work is what inspired us to write the third order non-principal term in the form of Eq 182 in the Appendix, after we had previously noted its stabilizing properties. These important works are complementary to our own work; they do not use continuous time approaches and tackle primarily the edge of stability problem or its subcases, while we focus on understanding gradient descent and applying that understanding broadly, including but not limited to the edge of stability phenomenon.

**Discrete models of gradient descent**. The desire to understand learning rate specific behavior in gradient descent has been a motivation in the construction of discrete time analyses. These analyses have provided great insights, from studying noise in the stochastic gradient descent setting (Liu et al., 2021; Ziyin et al., 2021b), the study of overparametrized neural models and their convergence (Gunasekar et al., 2018; Du et al., 2019; Allen-Zhu et al., 2019), providing examples when gradient descent can converge to local maxima (Ziyin et al., 2021a), the importance of width for proving convergence in deep linear networks (Du and Hu, 2019). We differ from these studies both in motivation and execution: we are looking for a continuous time flow which will increase the applicability of continuous time analysis of gradient descent. We do so by incorporating discretisation drift using BEA and showing that the resulting flow is a useful model of gradient descent, which captures instabilities and escape of local minima and saddle points.

**Understanding the difference between the negative gradient flow and gradient descent**. Elkabetz and Cohen (2021) recently examined the differences between gradient descent and the NGF in the deep learning context; their work examines the importance of the Hessian in determining when gradient descent follows the NGF. Their theoretical results show that neural networks are roughly convex and thus for reasonably sized learning rates one can expect that gradient descent follows the NGF flow closely. Their results complement ours and their approach might be extended to help us understand why the PF is sufficient to shed light on many instability observations in the neural network training.

**Second-order optimization.** By using second order information (or approximations thereof) to set the learning rate, DAL is related to second-order approaches used in deep learning. Many second-order methods can be seen as approximates of Newton's method $\boldsymbol{\theta}_t = \boldsymbol{\theta}_{t-1} - \nabla_{\boldsymbol{\theta}}^2 E^{-1}(\boldsymbol{\theta}_{t-1})\nabla_{\boldsymbol{\theta}}E(\boldsymbol{\theta}_{t-1})$. Since computing the inverse of Hessian can be prohibitively expensive for large models, many practical methods approximate it with tractable alternatives (Martens and Grosse, 2015). Foret et al. (2020) propose an optimisation scheme directly aimed at minimising sharpness, and show this can improve generalisation.

**Connection between drift and generalization**. We have made the connection between increased drift and increased generalization. This connection was first made by Barrett and Dherin (2021) through the IGR flow. Generalization has also been connected to the largest eigenvalue $\lambda_0$(Hochreiter and Schmidhuber, 1997; Keskar et al., 2016; Jastrzębski et al., 2018; Lewkowycz et al., 2020); recently Kaur et al. (2022) however showed a more complex picture, primarily in the context of stochastic gradient descent. The largest

eigenvalue could be a confounder to the drift as we have observed in Section 6.3; we hope that future work can deepen these connections.

## 9 Conclusion

We have expanded on previous works which used Backward Error Analysis in deep learning to find a new continuous time flow, called **the Principal Flow**, to analyze the behavior of gradient descent. Unlike existing flows, the principal flow operates in complex space which enables it to better capture the behavior of gradient descent compared to existing flows, including but not limited to instability and oscillatory behavior. We use the form of the Principal Flow to find new quantities relevant to the stability of gradient descent, and shed light on newly observed empirical phenomena, such as the edge of stability results. After understanding the core quantities connected to instabilities in deep learning we devised an automatic learning rate schedule, DAL, which exhibits stable training. We concluded by cementing the connection between large discretization drift and increased generalization performance. We ended by highlighting future work avenues including incorporating the principal flow in existing theoretical analyses of gradient descent which use the negative gradient flow, incorporating our understanding of the drift of gradient descent in other optimization approaches and specializing the PF for neural network function approximators.

**Acknowledgments**. We would like to thank the TMLR anonymous reviewers and the TMLR Action Editor for their useful feedback and comments. We would also like to thank Soham De and Michael Munn for discussions and feedback; and Frederic Besse, Marc Deisenroth, Patrick Cole, Shakir Mohamed and Timothy Lillicrap for their support.

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
