# OpenReview forum: "On a continuous time model of gradient descent dynamics and instability in deep learning"
_TMLR — Accepted by TMLR_

### Review · Reviewer_hnSw · 2022-08-22

**Summary Of Contributions:**

In this paper, the authors first proposes the Principal Flow (PF) as a continuous-time approximation for the discrete gradient descent method. The PF is derived from the approach of modified equation in numerical analysis, which is prevalent in the analysis of numerical methods for differential equations. The PF dynamics depends on the spectrum of the Hessian of the loss function. Compared with the traditional gradient flow (NGF), the PF changes the magnitude the update (right hand side of the dynamics) along each eigenvector direction of the Hessian by a logarithm function. The logarithm function makes the dynamics complex, and thus allows richer behaviors around stationary points.

After deriving the PF, the authors first studied the stability of the PF with respect to the eigenvalues of the Hessian. The results are parallel with that for GD. Showing the advantage of PF over NGF and IGR on the approximation of GD behaviors. Then, the PF is used to explain the behaviors of GD in different regimes observed in the edge of stability phenomenon. Extensive experiments show that the PF can indeed characterize GD behaviors in both the stable and edge of stability regimes.

Finally, the authors studied the closeness of GD and NGF, showing that the "discretization drift" is the crucial quantity that controls this closeness. From here, an algorithm which scales the learning rate using a quantity related with the discretization drift is proposed. Numerical results show that the proposed algorithm (DAL) can improve the stability of training (over GD). However, a tradeoff of stability and generalization on the other hand shows algorithms that depend less on the discretization drift generalize better.

**Broader Impact Concerns:**

The reviewer does not have concerns on the broader impact of the paper.

**Requested Changes:**

Questions are posted in the "strengths and weaknesses" part.

**Strengths And Weaknesses:**

Strengths:

1. The PF can well characterize the stability of GD. Compared with NGF and IGR, its extension to the complex space allows different stability behaviors of the dynamics around stationary points. This is very interesting and may potentially introduce more tools from continuous dynamics to study GD.

2. The paper provides extensive experimental evidence to justify the arguments made.

Weaknesses:

1. From the reviewer's perspective, the explanation to the edge of stability phenomenon in Section 5 is not enough. The explanations focus on the stability of GD on a quadratic landscape. However, on a quadratic landscape the GD cannot stay at the edge of stability---it either converges or diverges. Can the PF be used to explain why GD can stay at the edge of stability?

2. The contents in Section 6 seems not related with PF. Instead, the relation between the GD and NGF is studied. Could the authors provide some motivations from the PF for this section?

3. It is shown clearly that the PF characterizes the same stability of GD around a minimum. However, the paper does not provide a case in which the PF can help analyze the the behavior of GD that cannot be done by directly studying GD---it seems all results for PF in the paper can be obtained by simply look at GD. Can the PF extend our understanding for GD?

---

> ### Author Response · Authors · 2022-09-01
> **Thank you and discussion**
>
> We thank the reviewer for the detailed read, for the insightful comments and for the prompt review.  We would also like to thank the reviewer for highlighting the strengths of our work.
>
> We try to address the points that were highlighted by the reviewer as weaknesses below.
>
> > 1) From the reviewer's perspective, the explanation to the edge of stability phenomenon in Section 5 is not enough. The explanations focus on the stability of GD on a quadratic landscape. However, on a quadratic landscape the GD cannot stay at the edge of stability---it either converges or diverges. Can the PF be used to explain why GD can stay at the edge of stability?
>
> Can you please clarify what aspect of Section 5 made you think there is a limitation which assumes a quadratic landscape? We will try to clarify. Unlike for the quadratic case, the analysis in Section 5 is done for neural networks, where the Hessian changes between iterations; this what allows for the edge of stability to arise. Other results which show the PF can be used to approximate the behaviour of GD for non-quadratic functions in Figures 6, 7, 8 in the main paper and Figures 29 and 31 in the Appendix.
>
> When using the PF to approximate GD, the PF tells us how to approximate the behaviour of gradient descent given a Hessian landscape, and whether the PF resembles more the negative gradient flow or the positive gradient flow in a particular eigendirection,  depending on the corresponding eigenvalue. We confirm empirically that GD does indeed behave closer to the predicted flow by the PF in Figures 10 and 12 in the main paper.  The same figures show that when following the NGF the largest eigenvalue increases, following the positive gradient flow decreases it, and vice versa. This suggests that after reaching the 2/h threshold, following the PF leads to an alternating behaviour between increasing and decreasing the largest eigenvalue, above and below the line in continuous time (edge of stability behaviour). We also show that following the positive gradient flow can lead to an increase and then a decrease in the largest eigenvalue (Figure 10b) which again is consistent with the edge of stability.
>
> > 2) The contents in Section 6 seems not related with PF. Instead, the relation between the GD and NGF is studied. Could the authors provide some motivations from the PF for this section?
>
> The PF is what makes clear that $\nabla_{\theta}^2 E \nabla_{\theta}E$ is an important quantity which controls the drift - through the series we constructed in Section 3. From there we looked at  $\nabla_{\theta}^2 E \nabla_{\theta}E$ more closely and found that it can be used to capture not only the difference between the PF and the NGF, but also the difference between the PF and GD.

---

> ### Author Response · Authors · 2022-09-01
> **Thank you and discussion (2)**
>
> > 3) It is shown clearly that the PF characterizes the same stability of GD around a minimum. However, the paper does not provide a case in which the PF can help analyze the the behavior of GD that cannot be done by directly studying GD---it seems all results for PF in the paper can be obtained by simply look at GD. Can the PF extend our understanding for GD?
>
> We believe that the value of the PF compared to other analysis of GD is as follows (we detail each of them below):
>   * a) The PF provides one encompassing framework to understand GD and allowed us to find previously unknown insights about GD, both around and outside local minima.
>   * b) The PF better captures GD, it can be used to improve existing results using continuous time analysis of GD, by replacing the NGF with the PF.
>   * c) The PF expands the use of BEA in deep learning and sheds light on existing BEA flows.
>
>
> Detailed explanations for each point above:
>
> a) The PF can be used to analyse GD across 1 iteration or multiple iterations, around or outside a local minima and saddle points. While individual discrete analyses might provide similar insights for individual cases discussed, the PF provides an encompassing tool which can be applied generally. The PF allowed us to find previously not known insights about gradient descent. For instance, through the PF we have found the importance of the stability coefficient in understanding gradient descent. The stability coefficient of each eigendirection shows both the importance of the eigenvalues of the Hessian, but also the importance of the dot product between the gradient and the Hessian eigenvectors. The stability coefficient can be used to understand instabilities of gradient descent, and while the edge of stability paper focused solely on the largest eigenvalue, we were able to show that this is not sufficient to understand instabilities. Through the continuous time perspective provided by the PF, by measuring the gap between the NGF and GD, we were able to devise an approach to approximate discretisation drift and use it to stabilise training.
>
> b) The biggest challenge of continuous time analysis is the gap between the NGF and gradient descent. By decreasing the gap through the PF, continuous time analysis can be expanded to be closer to GD. Recent works have shown how discrete updates break symmetries present in continuous time flows used to describe gradient descent - it will be interesting to expand on these works and observe whether the PF still exhibits those symmetries observed in existing methods or not. The application of the PF can also be expanded beyond supervised learning - in games stability analysis of continuous time flows has been used to devise explicit regularisers to ensure convergence around a Nash equilibrium; using the PF instead of the NGF can lead to regularisers which work better in practice.
>
> c) The PF also expands the use of backward error analysis (BEA) in deep learning. We hope that using this approach, new flows can be developed, including more non-principal terms. It also sheds light on existing results obtained using BEA, such as the modified loss of implicit gradient regularisation (described in the paper via the IGR flow). The implicit regularisation effect of the IGR flow has been used to show why larger learning rates converge to minima which generalise more. The PF suggests that the IGR flow is not always a good model of gradient descent training (specifically, it is not a good model around edge of stability areas) - this is a novel result we present. Finding *when* in training the IGR effect is strongest is important as it can help find when in training it is sufficient to add explicit regularisation in order to increase generalisation, and increase speed of training.

---

> > ### Author Response · Authors · 2022-09-21
> > **Update on comment visibility**
> >
> > Dear reviewer,
> >
> > We have only realised that our second comment which served as a reply might not have been visible to you. We apologise and we have now mitigated this issue.

---

> > > ### Author Response · Authors · 2022-09-28
> > > **Updates to the manuscript**
> > >
> > > Dear reviewer,
> > >
> > > We have now updated the manuscript to include changes to Section 5 (on the edge of stability connection) and Section 6 (the connection between the PF and DAL) based on your comments and suggestions. The changes made are highlighted in orange to make them easier to find.
> > >
> > > For Section 5, we added a new set of experiments where we compare the model of gradient descent at the edge of stability area provided by the PF against existing flows as well as comment on how we use the PF to model the edge of stability, in the last paragraph of that section. For Section 6, we further highlighted the motivation of DAL starting from the PF.
> > >
> > > Thank you,
> > > the authors.

---

> > > > ### Comment · Reviewer_hnSw · 2022-10-12
> > > > **Thanks for the response**
> > > >
> > > > Thanks for the detailed response to my questions, as well as the effort to improve the paper. Most of my questions have been addressed. The PF flow is a good choice if one wants to study GD from a continuous perspective. The only concern is still the lack of an example where the PF flow can provide much more insights than analyzing the discrete GD scheme.

---

> > > > > ### Author Response · Authors · 2022-10-13
> > > > > **Thank you and follow up**
> > > > >
> > > > > We thank the reviewer for the useful comments and for helping us improve the manuscript.
> > > > >
> > > > > In terms of novel insights coming from the PF, we provide a few examples:
> > > > >   * Using the PF we found an important novel quantity that determines the behavior of gradient descent in the stability coefficient (Definition 3.4). We confirm empirically the significance of the stability coefficient in Figure 10 which shows that model training can be stable when $\nabla_{\theta} E ^T u_0$ is close to 0 even $\lambda_0 >2/h$. Similarly, we show that a higher value of the stability coefficient is associated with higher instabilities (Figures 10, 11, 12 in the main paper and Figures 34, 35, 36 in the Appendix).
> > > > >   * By asking when the PF is close to the NGF, we developed DAL and its variants which we used to provide a stable training approach.
> > > > >   * By understanding when the PF and the NGF differ and when they lead to similar trajectories, we can understand the trajectory of the largest eigenvalue of the Hessian $\lambda_0$ or the loss in gradient descent, as we show in Figure 13.
> > > > >   * Through analysing the PF we were able to construct a setting which enabled us to show that one eigendirection is sufficient to cause instability and changes in $\lambda_0$ (Figure 14).
> > > > >   * Through an analysis of the PF and non-principal terms we can show which components of GD can cause stability and which instability (Figure 6).
> > > > >
> > > > > One of our main goals is to establish a new continuous time model of gradient descent, and show it is a good model in the neural network context. Once that is established through this work, the PF can be integrated into existing analyses which do not have a discrete time counterpart. One such example could be the Neural Tangent Kernel, which uses the NGF to analyse the convergence properties of infinitely wide NNs. By integrating the PF into the Neural Tangent Kernel one can better understand its limitations and underlying assumptions.

---

### Review · Reviewer_uCdx · 2022-09-02

**Summary Of Contributions:**

This paper proposes a new continuous time flow to approximate gradient descent. Specifically, based on the Backward Error Analysis (BEA) technique in numerical analysis, the authors propose the principle flow (PF) with higher order error terms. Based on the principle flow, the authors are able to explain divergent and oscillatory behaviors in deep learning training, and propose a learning rate adaptation method to reduce instabilities.

**Broader Impact Concerns:**

No concerns.

**Requested Changes:**

Please refer to the part “Weakness”. Since the conclusions of this paper heavily based on the above analysis, I hope the authors can comment on this.

**Strengths And Weaknesses:**

trengths:
	The idea using BEA to provide a more precise continuous flow for (S)GD is interesting.
	The paper is well-written.

Weakness:
I find a technical flaw which may severely deteriorate the contribution of this paper.
From Eq. (11) to Eq. (12) (or the proof of Theorem 13), the authors replace \sum_{p=0}^\infty {- 1/(p+1) * z^p}  with log(1-z)/z, which is not reasonable to me. A simple example is that the former term diverges to -\infty when z>1 while the latter is a finite complex number (corresponding to the case λ_i>1/h in page 7.  All of the divergent and oscillatory analyses are based on this case). However, the latter term yields a complex number when z>1 (which is the focus of this paper). In a word, log⁡(1-z)/z is only a proper approximation for the case |z|≤1, but the authors use the approximation for all the cases, and erroneously arrived at the conclusion “for the case λ_i>1/h, PF is a complex flow” (to me, it seems PF even cannot be defined in this case).

---

> ### Author Response · Authors · 2022-09-02
> **The PF and BEA**
>
> Thank you for the review.
>
> We believe there is a misunderstanding in what BEA provides. In Section 2.1, which describes BEA, clarifies that BEA provides the *Taylor expansion* of a modified vector field at $h=0$. We specifically note after Eq. 6 that the series provided by BEA is usually divergent, but we can use it to find a function whose Taylor expansion in h leads to the series (Eq 7).
>
> Regarding the PF: we do not derive the PF (Eq. 12, which in Def 3.1) from the BEA series (Eq. 11). Our logic is the reverse. We used Thm 3.2 to show that the BEA series can be obtained from the PF by a Taylor's expansion at $h=0$, and thus the PF defines the modified vector field given by BEA (Eq. 7). We never claimed that the series and the PF are the same, namely, Eq. 12 is an analytical continuation of Eq. 11 outside of its convergence radius. We will make this point clearer in the paper.
>
> We also note that in the quadratic case, the PF tracks the behaviour of GD exactly, and we show this consistently in Figures 2 and 5 in the main paper, even when  $\lambda_0 > 1/h$ or even in the case of divergent behaviour, when  $\lambda_0 > 2/h$.

---

> > ### Comment · Reviewer_uCdx · 2022-09-09
> > **Response**
> >
> > I would like to thank the authors for the reply. My concern remains, however. Even though the authors claim that “Our logic is the reverse.” and “We used Thm 3.2 to show that the BEA series can be obtained from the PF by a Taylor's expansion at $h=0$” (meanwhile, the organization of current paper seems to indicate that BEA indicates PF, as BEA is explored by a lot before PF is defined. I suggest the author to reorganize the paper if the logic is reverse), the BEA series serves as the only explanation for why PF is defined as this way. The rest of the paper verifies that PF is a reasonable approximation by showing that it shares similar properties with GD – which is post-hoc. Therefore, I would expect the approximation error is provably small in all of the case considered in this paper -- rather than at $h=0$, which is quite restricted. As I pointed out in the previous review, for the case $h>\frac{1}{\lambda}$, which is the focus of this paper, PF defers significantly with BEA series and BEA series can thus hardly support the definition of PF in this case. Please let me know if I have missed anything.

---

> > > ### Author Response · Authors · 2022-09-21
> > > **Clarificaion on BEA usage**
> > >
> > > We thank the reviewer for the comments. We reply to comments inline and will adjust the manuscript to further clarify the points below based on your review.
> > >
> > > > meanwhile, the organization of current paper seems to indicate that BEA indicates PF, as BEA is explored by a lot before PF is defined.
> > >
> > >
> > > Without developing the BEA series first, we would not know the form of the PF. The PF is the h-dependent vector field whose Taylor's series at $h=0$ coincides with the principal terms of the asymptotic series prescribed by the BEA. Thus to know the PF, we need to know the BEA series first. This is why we start with BEA and then proceed to find the PF.
> > >
> > > Given the reviewer's comments, we understand that our logic needs be clarified and we will change the manuscript to reflect that in the coming days.
> > >
> > > > I would expect the approximation error is provably small in all of the case considered in this paper -- rather than at $h=0$, which is quite restricted.
> > >
> > > The mathematical results in the paper are not restricted to h=0.  BEA does not assume an infinitely small learning rate (i.e., h=0), as it is common with other continuous time analysis methods. For a given **finite** learning rate h, BEA provides a modified vector field with local error of order $\mathcal{O}(h^p)$ for each p.
> > >
> > > Providing a rigorous estimate of the error bound between GD and PF in general is a very hard problem which is likely to depend on the neural net architecture through the form of its Hessian (because of the non-principal terms). We note that for BEA in general - independent of the approach taken here - it is notoriously hard to find rigorous estimates which prove that adding an extra term in the BEA series decreases the absolute value of the error. That is, we cannot generally state that going from an order $\mathcal{O}(h^p)$ to  $\mathcal{O}(h^{p+1})$ will decrease the absolute value of the error. We refer the reviewer to Thm 7.6 in Chapter IX.7.3 in the Geometric Numerical Integration book by Hairer et al for a more general discussion.
> > >
> > >
> > > > As I pointed out in the previous review, for the case $h>\frac{1}{\lambda}$, which is the focus of this paper, PF defers significantly with BEA series and BEA series can thus hardly support the definition of PF in this case.
> > >
> > > We use BEA to find the PF such that it corresponds to the asymptotic series prescribed by the BEA. Once we have the PF, we investigate its properties. We first show that for the quadratic case the PF matches the trajectory of GD exactly regardless of the learning rate (Remark 3.1) - we prove this independently of BEA in A.6. This applies regardless of the relationship between h and \lambda. Beyond the quadratic case, we show how the PF explains empirically observed behaviours around critical points and beyond (Section 3.2 and more). We also observe empirically that the local error between GD and PF is smaller than previous proposed continuous flows, including for NNs (Figure 7), also in situations where $h > 1/\lambda_0$ and the PF is complex. We show more broadly in Figure 8 how the PF can be used to predict behaviours of large scale NNs which cannot be explained by existing flows.

---

> > > > ### Author Response · Authors · 2022-09-26
> > > > **Updated manuscript**
> > > >
> > > > Based on the reviewer's comments, we have updated the manuscript to clarify changes around the definition of the PF and BEA. The changes are highlighted in blue in Section 3 for your convenience. Thank you!

---

> > > > > ### Comment · Reviewer_uCdx · 2022-12-04
> > > > > **Thank you for the response and updated manuscript**
> > > > >
> > > > > My concern about the paper is whether the approximation error can be bounded for the main cases the paper studied. In the updated version, for quadratic functions, authors provided the approximation analysis for general h and lambda. It is a good improvement. I suggest authors make further discussions in the paper for the non-convex DNNs. My concern is alleviated. Thank you.

---

> > > > > > ### Author Response · Authors · 2022-12-13
> > > > > > **Thank you and manuscript update**
> > > > > >
> > > > > > We thank the reviewer for the comment and their thoughts on the approximation error.
> > > > > >
> > > > > > We have added on page 7 a paragraph further discussing this topic (highlighted in blue), providing clarifying information as well as connecting to the Fundamental theorem which could be expanded to provide formal bounds.

---

### Review · Reviewer_Psao · 2022-09-18

**Summary Of Contributions:**

The paper proposes a continuous-time flow approximation to the gradient descent algorithm.

The proposed approximation is found to approximate the actual gradient descent much better than the existing flow methods.

The paper then proposes the DAL, a second-order method for setting the learning rate to improve the stability of training.

**Broader Impact Concerns:**

Fine.

**Requested Changes:**

While I feel that the scientific content of the paper is decent, the discussion of the motivation and the related works needs extensive updates for me to recommend acceptance.

1. Extensive discussion of why continuous-time approximation is needed, especially given many works exist to study the original discretized version. Especially, the authors need to point out clearly, in light of the previous works, what the previous works on discrete-time gradient descent cannot offer and the continuous-time theories excel at. I do personally believe that continuous-time theories are important and have a lot of merits -- but the authors are responsible for explaining why. To my knowledge, the following works are worth discussing (saddle points, stability, etc).
- https://proceedings.mlr.press/v139/liu21ad.html
- https://openreview.net/forum?id=SkNksoRctQ
- https://openreview.net/forum?id=uorVGbWV5sw
- https://openreview.net/forum?id=9XhPLAjjRB

2. In fact, the authors did not mention any work that studies gradient descent in discrete-time, as if they do not exist. Outside the references I point out above, the authors should look for more works in this direction and discuss



**Strengths And Weaknesses:**

Strength: the strength of the paper is the novelty of the proposed approximation and the extensive experiments used to validate its empirical relevance.

Weakness:
1. I feel that there is not any new significant insight the new theory provides. The theory mostly confirms previous experimental discoveries and the insights the previous works have provided. If theory is really novel and meaningful, it should also provide some new insight that no previous work has discovered.

2. The proposed DAL algorithm also does not feel very novel. It can also be obtained by a simple application of Newton's method.

3. This is my major criticism. I do not feel that the paper is sufficiently motivated. The paper has glossed over the discussion of a very important motivation: why is a continuous-time approximation needed in the first place?
- For example, the instability threshold 2/h can be directly understood from a discrete-time treatment (in fact, even more precise treatments have been established). The DAL method also does not require a continuous-time model to justify. See the requested changes section for more detail

4. A different question -- gradient flow is known to work very well for NTKs, does the present work offer a better understanding for NTKs when trained at a larger learning rate?

---

> ### Author Response · Authors · 2022-09-23
> **Changes to the manuscript**
>
> We would like to thank the reviewer for the comments and for providing additional insight and connections.
>
> We start with highlighting the requested changes, which have been incorporated in the manuscript (shown in red in the pdf):
>  * Based on the reviewer’s comments we noticed that the exposition of our results can be improved. We thus rewrote the bullet points in the introduction to highlight both the motivation to use continuous time and our contributions.
> * We have added a subsection in Section 2 which further motivates the use of continuous time approaches and restructured the section to further showcase our motivations.
> * In the same section, we have now highlighted the difference between discrete and continuous methods.
> * We have added a subsection in the related work to discuss discrete time methods (Section 8).

---

> > ### Author Response · Authors · 2022-09-23
> > **Regarding insights and DAL**
> >
> > > I feel that there is not any new significant insight the new theory provides.
> >
> > We believe that the proposed flow provides novel insights. Specifically we show the importance of complex flows in understanding gradient descent in continuous time, something that had not been previously done. We also found that an important quantity that determines the behavior of gradient descent is the stability coefficient (Definition 3.4). While the edge of stability paper (Gradient Descent on Neural Networks Typically Occurs at the Edge of Stability, Cohen et al) focused primarily on the value of the largest eigenvalue $\lambda_0$, we show the significance of the stability coefficient $\log(1-h \lambda_i)/ (h \lambda_i) \nabla_{\theta} E ^T u_i$ . Indeed, we  confirm empirically in Figure 10 that model training can be stable when $\nabla_{\theta} E ^T u_0$ is close to 0 even $\lambda_0 >2/h$ Similarly, we show that a higher value of the stability coefficient is associated with higher instabilities (Figures 10, 11, 12 in the main paper and Figures 34, 35, 36 in the Appendix).
> >
> > We are also the first, to our knowledge, to show that discretisation drift can be approximated rather cheaply and we verify this empirically (Section 6, and Figures 15 and 16). We then use this to show that we can control discretisation drift (via DAL) and that this can be incorporated into other optimization algorithms. Furthermore, we find a clear criterium ($\nabla_{\theta}^2 E \nabla_{\theta} E = 0$ ) to indicate when gradient descent follows the exact same path as the negative gradient flow. This is relevant as their closeness early in NN training has been previously observed.
> >
> > We also note that the PF can lead to further theoretical work, as the reviewer has highlighted. This includes applications to NTK but also expansions on other works which previously used the NGF as a continuous-time model of gradient descent. The aim of this work was to introduce the PF and show it can be useful in gradient descent analysis and for understanding NN training. We hope that once that is done follow up work can build on our approach. We have further highlighted this in the manuscript.
> >
> > > The proposed DAL algorithm also does not feel very novel. It can also be obtained by a simple application of Newton's method.
> >
> > We would like to highlight that the aim of DAL is to control the gradient descent and thus the trade-off between stability and generalisation. In doing so, it does provide a novel way to incorporate second order information, but it is different from using Newton’s algorithm (or approximations thereof) in a few ways:
> >   *  The Newton algorithm uses the update $H^{-1}g = H^{-1} \hat{g} ||g||$. DAL uses the update $g/||H \hat{g}||$ . While these updates have some similarities in their form,  the core difference between DAL and Newton is whether $H$ or $H^{-1}$ gets applied to the unit normalised gradient. Indeed, even for a diagonal $H$ with diagonal entries $\lambda_1, \lambda_2, ... \lambda_n$ we have that $H^{-1}\hat{g} = \sum_i  \frac{1}{ \lambda_i} \hat{g}_i$ which is different to what DAL does $\frac{1}{\sum_i  \lambda_i \hat{g}_i}$. The reviewer might have also been referring to obtaining a learning rate from a Taylor expansion of the loss E (as Newton is derived) but instead of computing the optimal direction of change, computing the optimal learning rate for a gradient descent algorithm. In that case, the optimal learning rate becomes $1/  \hat{g} ^T H  \hat{g}$, which is related but not the same as DAL.
> >   * An important aspect of our approach is that because it is derived from discretisation drift, we can use it in a variety of settings. We have shown in the paper how DAL can be adapted to momentum: the main idea is to increase the contribution of local gradients to the momentum moving average when the drift is large, and decrease it otherwise. We can do so only through our understanding of drift. Similarly, while we focus on using one learning rate in the paper (and thus use the norm operator in DAL), this is not necessary. We can construct per parameter learning rates and show preliminary results in the Appendix (Figures 43 and 44).
> >   * The aim of DAL is to control the discretisation drift of gradient descent and show that this can affect both the stability and performance of gradient descent, through DAL-p. Beyond providing a training tool, DAL-p is another tool to use to understand gradient descent. Importantly, the ability to control the drift also enhances the generalisation capabilities of the method, while Newton like methods tend to overfit and do not have mechanisms allowing for a control of the generalisation v.s. stability trade-off. The only reason we can do this is through the understanding of the drift (higher drift leads to higher generalisation).

---

> > > ### Author Response · Authors · 2022-09-23
> > > **Motivation, NTK and other connections**
> > >
> > > >  This is my major criticism. I do not feel that the paper is sufficiently motivated. The paper has glossed over the discussion of a very important motivation: why is a continuous-time approximation needed in the first place?
> > >
> > > We hope this has been addressed by the changes in Section 2, which highlight the importance of continuous time flows.
> > >
> > > > A different question -- gradient flow is known to work very well for NTKs, does the present work offer a better understanding for NTKs when trained at a larger learning rate?
> > >
> > >
> > > Thank you for highlighting this. Yes, we do believe that looking at NTK using the PF can be a fruitful endeavour. The NTK paper (Neural Tangent Kernel: Convergence and Generalization in Neural Networks by Jacot et al) shows that in the infinitely wide limit gradient descent for NNs follows Kernel gradient descent which converges, and uses the NGF as the approximation to ANN gradient descent. Thus if in their approach we wanted to understand the effect of the learning rate, one can replace the NGF with the PF in their work -- specifically the contents of Thm 2 in the NTK paper and aim to find conditions under which convergence occurs. Another approach of integrating the PF analysis into the NTK discussion is looking to see if for wide neural networks the Hessian has properties such that it makes it less likely that the unstable regimes of the PF are reached, especially considering existing works have highlighted limitations of the NTK approach in the finite width regime [a, b], and specifically in the high learning rate regimes, as [b] states “weight decay and the use of a large learning rate break the correspondence between finite and infinite networks”. We have now added this in the future work paragraph.
> > >
> > > [a] Limitations of the NTK for Understanding Generalization in Deep Learning, Vyas et al.
> > >
> > > [b] Finite Versus Infinite Neural Networks: an Empirical Study Lee et al.
> > >
> > >
> > > > Extensive discussion of why continuous-time approximation is needed, especially given many works exist to study the original discretized version.
> > >
> > > We have now expanded Section 2 to include additional advantages of continuous time methods. We have also included a section on related work on discrete time and referenced the works provided by the reviewer.

---

> > > > ### Comment · Reviewer_Psao · 2022-10-02
> > > > **Thanks for the reply**
> > > >
> > > > I am satisfied with the efforts that the authors put into improving the draft.
> > > >
> > > > A minor comment: some typographical errors still exist. For example, there are places where the authors should use \citep instead of \cite. Please fix these points.

---

> > > > > ### Author Response · Authors · 2022-10-03
> > > > > **Thank you**
> > > > >
> > > > > We thank the reviewer for their comments which improved the manuscript and for appreciating our work.
> > > > >
> > > > > We will do a detail read and fix existing typographical errors.

---

### Decision · Action_Editors · 2022-12-11

**Recommendation:** Accept with minor revision

**Comment:**

The article proposes a continuous time model to understand the behavior of gradient descent.
There was a healthy amount of discussion following the initial reviews. During the discussion, several perceived shortcomings and technical aspects could be clarified. The authors made various updates to the manuscript in response to the discussion with the reviewers.

Overall, I find the article discusses a topic that is of interest to the TMLR audience. While the final reviewer recommendations are mixed and there are still some aspects that could be further improved, after evaluating the discussion, rebuttal, article updates, and discussing with the reviewers, I find that the merits of this article outweigh the weaknesses and that the paper meets the bar for acceptance.

I am recommending accept with minor revision and ask the authors to take one final pass over the article taking the final reviewer comments into account. In particular, uCdx indicated that the technical flaws they perceived initially have been resolved for the case of quadratic functions, but that for general cases they were not fully addressed. Reviewer hnSw indicated that the article proposes a good continuous approximation for GD, but that they would expect to see new insights. The authors gave a few examples in the forum later. Reviewer Psao was most favorable after the discussion, but also had minor final suggestions.


**Audience:**

Yes, I believe a good fraction of individuals in TMLR's audience would be interested in this paper.

**Claims And Evidence:**

The article provides a good amount of background. Theoretical statements are accompanied by well organised proofs in the appendix. The revisions during the discussion period improved motivation and could clarify some technical aspects. The article includes a good amount of experiments in support of their claims. A final revision should still make efforts to address the final comments from the reviewers, particularly of uCdx.

---

> ### Author Response · Authors · 2022-12-20
> **Thank you and manuscript updates**
>
> We thank the AE for their deep consideration of the paper and for their appreciation of our results.
>
> We have updated the manuscript to include revisions the AE has highlighted, including:
>   * On page 7 adding a discussion regarding error bounds of the PF in the non-convex case, as requested by reviewer uCdx and highlighted by the AE. The changes done at the behest of  reviewer uCdx are highlighted in blue.
>    * We have highlighted the new insights as requested by reviewer hnSw using the following (the changes done at the behest of reviewer hnSw are highlighted in orange):  We have provided captions which better highlight our results in Figures 11, 12, 13, 15.
> At the end of Section 4 we have connected the stability analysis results with observations from training neural networks, including gradient descent being repelled around non-strict saddles. This includes an additional experiment with supporting evidence.
> At the end of Sections 5 and 6 we have provided additional paragraphs to highlight the novelty induced by that section.
>  * For reviewer Psao18, we have added additional explanations around DAL and connections to the PF in Section 6 and further connections from the discussion have also been added in the Supplementary Material (Section A.7.3, highlighted in red). Based on their original comments we have also added a  paragraph titled “Why not more instability?” in Section 5.  We have also added figures showing the landscapes learned with DAL, Figures 46-50 in the Appendix.
>
> We will do another detailed pass on the manuscript following the above changes.

---

> > ### Author Response · Authors · 2023-01-19
> > **Upload of camera ready version**
> >
> > Dear AE,
> >
> > We thank you and the reviewers again for the useful comments and insights. We have incorporated the comments as well as done additional passes through the manuscript, and uploaded the camera ready version.